# GFSE: A FOUNDATIONAL MODEL FOR GRAPH STRUCTURAL ENCODING

## ABSTRACT

Foundation models have recently shown remarkable promise by leveraging extensive pre-training on diverse datasets to acquire generalizable representations, which enable effective transfer to a wide range of downstream tasks. In the graph domain, however, most existing pre-training models are tailored to specific domains, primarily due to the inherent differences in semantic meanings of graph features across various contexts. Additionally, most existing models struggle to capture the rich topological complexity of graph structures, leading to inadequate exploration of the embedding space. To address these challenges, we propose a novel **G**raph **F**oundational **S**tructural **E**ncoder (**GFSE**) that identifies universal structural patterns, facilitating a unified feature embedding space suitable for diverse domains, including molecular structures, social networks, and citation networks. GFSE is the first cross-domain graph structural encoder pre-trained with multiple self-supervised learning objectives. Built on a Graph Transformer, GFSE incorporates attention mechanisms biased by graph structural information, allowing it to encode intricate multi-level and fine-grained topological features within complex graph structures. The pre-trained GFSE produces generic and theoretically expressive positional and structural encoding for graphs, which can be seamlessly integrated with various downstream graph feature encoders, including graph neural networks for graphs with vectorized features and Large Language Models for text-attributed graphs. Comprehensive experiments on synthetic and real-world datasets demonstrate GFSE's capability to significantly enhance the model's performance while requiring substantially less task-specific fine-tuning. Notably, GFSE boosts the performance by an average margin of $20.48\%$ across eight real-world datasets, highlighting its potential as a powerful and adaptable foundational encoder for graph-structured data.

## 1 INTRODUCTION

Foundation models are recently attracting a surge of interest in natural language processing (Achiam et al., 2023; Bubeck et al., 2023; Touvron et al., 2023), computer vision (Radford et al., 2021; Ramesh et al., 2021), audio (Yang et al., 2023; Borsos et al., 2023), etc. However, the application of such models in the graph domain remains relatively under-explored. Due to the inherent difference in dataset-specific features, most prior graph pre-training models are specialized for certain areas, such as molecules (Zhang et al., 2020; Sypetkowski et al., 2024), proteins (Nijkamp et al., 2023), and knowledge graphs (Galkin et al., 2023). These specialized models require domain-specific knowledge and suffer from limited transferability to different graph domains. Recent efforts attempt to harness LLMs to unify feature spaces of different graph domains using text (Chen et al., 2024b; Tang et al., 2023; Liu et al., 2023a; Kong et al., 2024). However, the text-based representations used by LLMs inherently lose the rich structural information encoded in the graph structure, leading to unsatisfactory performance on graph learning tasks (Fatemi et al., 2023; Zhao et al., 2023; Wang et al., 2024).

To advance the applicability of graph pre-trained models across diverse domains, we propose a paradigm shift that emphasizes the inherent structural patterns within graphs as universal attributes that are domain-agnostic and not tied to specific datasets. These patterns facilitate knowledge transfer across downstream tasks and datasets. For instance, social networks commonly exhibit small-world properties and community structures, while biological networks reveal recurring motifs and hierarchical modularity. Similarly, citation networks and the World Wide Web share characteristics

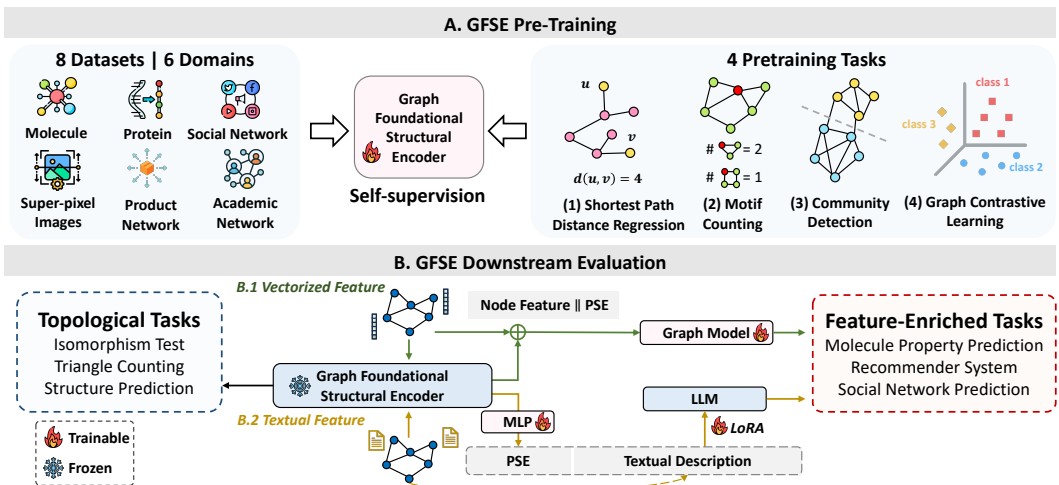

Figure 1: A) GFSE is pre-trained on 8 datasets from 6 different domains. Pre-training tasks include *reconstruction* (shortest path distance regression, motif counting) and *contrastive learning* (community detection, graph contrastive learning). B) GFSE generates generic and expressive Positional and Structural Encoding (PSE) to tackle topological tasks. GFSE can also be seamlessly integrated into downstream feature encoders for feature-enriched tasks by concatenating with initial vectorized features or prepending the generated PSE to the textual description as a soft token.

like scale-free degree distributions and core-periphery structures. However, developing a graph pre-trained model that can capture these diverse structural patterns in a generalizable manner, while remaining applicable enough to handle domain-specific adaptation, presents significant challenges. These challenges highlight the need for innovative pre-training strategies that focus on universal graph characteristics rather than domain-specific features.

**Proposed work**. To address the challenges of cross-domain pre-training and effectively capturing universal structural encoding, we propose **GFSE**, a **G**raph **F**oundational **S**tructural **E**ncoder, as shown in Figure 1. GFSE is pre-trained across diverse graph domains using multiple self-supervised pre-training tasks, including shortest path distance regression, motif counting, local community detection, and graph-level contrastive learning. Each pre-training task targets a critical and necessary aspect of graph structure, enabling GFSE to capture a comprehensive understanding of graph topology. GFSE employs a Graph Transformer enhanced with biased attention mechanisms. Notably, the relative positional encoding, derived from the random walk matrix, is explicitly integrated into the attention bias term. This design allows GFSE to effectively capture intricate structural dependencies among node pairs during pre-training, ensuring both efficiency and theoretically guaranteed expressiveness. GFSE's versatility extends to various graph learning scenarios. The pre-trained GFSE can produce generic and expressive Positional and Structural Encodings (PSE) for topological tasks. In feature-enriched contexts, the generated PSE can seamlessly augment vectorized features or integrate with text encoders (*e.g.,* LLMs) for text-attributed graphs. This applicability enables GFSE to serve as a powerful component in any graph foundational model.

The **contributions** of this work: (1) We propose GFSE, the first cross-domain graph structural encoder pre-trained with four essential self-supervised learning objectives. Extensive experiments show the effectiveness of these pre-training tasks, leading to an average performance improvement of $20.48\%$ across eight real-world datasets on downstream graph models. (2) We provide theoretical justification and empirical results demonstrating GFSE's ability to generate expressive PSE. (3) GFSE serves as a plug-and-play solution for any graph foundational model to incorporate structural information. By focusing on universal structural patterns, GFSE paves the way for more generalizable and adaptable graph encoding, potentially reducing the need for domain-specific pre-training in many applications.

## 2 RELATED WORK

**Graph Pre-training**. Graph self-supervised learning approaches are typically pre-training graph models, *e.g.,* GNNs or Graph Transformers, on a massive amount of labeled graphs with inherent features by reconstructing the structures or masked attributes (Cui et al., 2020; Hou et al., 2022; Kipf

& Welling, 2016b; Hu et al., 2020b; Wang et al., 2017; Xia et al., 2024; Xia & Huang, 2024; Zhao et al., 2024b; Mizera et al., 2024). Some works also utilize contrastive learning to enhance node and graph-level representation learning (Han et al., 2022; Hassani & Khasahmadi, 2020; Velickovic et al., 2019; Hu et al., 2019; Lee et al., 2022; Li et al., 2021; Lu et al., 2021; Sun et al., 2019; 2021; Xu et al., 2021a; Galkin et al., 2023; Zhao et al., 2024a). These methods, while effective in certain domains, exhibit limited generalizability across different graph domains due to their tailored design for specific types of data. Additionally, there have been graph prompt techniques (Huang et al., 2024; Fang et al., 2024) that can be used to enhance model adaptation over graphs. There have also been some attempts at cross-domain graph pre-training models (Qiu et al., 2020; Davies et al., 2023). Unfortunately, all these models rely on one singular pre-training task (*i.e.,* contrastive learning), and usually fail to capture fine-grained structural features at node level or edge level (Mao et al., 2024).

**LLM-based Graph Foundation Model**. With the success of foundation models in the NLP realm, recent efforts also harness LLMs to develop domain-specific graph foundation models by flattening graph structures and associated textural information into prompts (Chen et al., 2024a; Tang et al., 2023; Ye et al., 2023; Qian et al., 2023; Zhao et al., 2023; Guo et al., 2023; Chen et al., 2024b; Liu et al., 2023a; Kong et al., 2024; Chen et al., 2024c; Fan et al., 2024; Zhang et al., 2024; Li et al., 2024). Nevertheless, recent studies show that LLM demonstrates an unsatisfying ability to reason and understand complicated structures within graph (Fatemi et al., 2023; Zhao et al., 2023; Wang et al., 2024). In this work, we focus on developing a foundation model dedicated to encoding the rich topological information, without relying on associated text. Our approach complements LLMs on text-attributed graphs, serving as a foundational structural encoder for general graphs.

**Positional and Structural Encoding (PSE)**. Traditional PSEs include hand-crafted features such as Laplacian PE (Davies et al., 2024; Kreuzer et al., 2021; Beaini et al., 2021; Wang et al., 2022), shortest-path distance (Li et al., 2020; Ying et al., 2021), kernel distance (Mialon et al., 2021), random-walk encoding (Ma et al., 2023; Dwivedi et al., 2021; Brüel-Gabrielsson et al., 2022; Rampášek et al., 2022), node degree centrality (Ying et al., 2021), *etc*. Some studies have introduced specialized networks designed to adaptively learn PSE to enhance performance (Kreuzer et al., 2021; Dwivedi et al., 2021; Chen et al., 2022; Lim et al., 2022). GPSE (Liu et al., 2023b) proposes to pre-train a positional and structural encoder on domain-specific graphs to generate PSE. However, GPSE still suffers from limited transferability and expressiveness across other domains, due to its simplistic pre-training backbone and randomized node features. Consequently, the effectiveness of GPSE varies with specific tasks and graph models. Finding the most effective and versatile PSE remains an open challenge that requires further innovation.

## 3 PROPOSED METHOD

As shown in Figure 1, we collect graph pre-training datasets from six different domains, including molecules, proteins, social networks, images, product networks, and academic networks. GFSE utilizes a transformer-based architecture with biased attention to incorporate relative inductive bias within graph structures (Sec. 3.1). GFSE is pre-trained with four challenging self-supervision tasks simultaneously, each designed to enhance a crucial aspect of structural awareness and promote encoding quality (Sec. 3.2). GFSE generates expressive positional and structural encoding (PSE) for topological tasks. Moreover, the generated PSE can be seamlessly integrated into graphs with vectorized features or textual features, to enhance the downstream performance (Sec. 3.3).

### 3.1 ARCHITECTURE

Previous work (Liu et al., 2023b) uses randomized features to replace initial node features. However, it leads to poor generalizability across different domains. In this work, we propose to use both absolute and relative random-walk positional encoding as the initial features. Formally, let $G(V, E)$ represent an input graph, where $V$ and $E$ denote the set of nodes and edges, respectively. $\mathbf{A} \in \mathbb{R}^{N \times N}$ indicates the adjacency matrix, where $N$ is the number of nodes, and $\mathbf{D}$ is the degree matrix. Random Walk matrix is defined as $\mathbf{M} = \mathbf{D}^{-1}\mathbf{A}$, where $\mathbf{M}_{i,j}$ indicates the transition probability from the $i$-th node to the $j$-th node. Following previous works on random walk encoding (Ma et al., 2023), we calculate the $d$-dimensional encoding for each node and all node pairs.

$$\mathbf{P}_i = [\mathbf{I}, \mathbf{M}, \mathbf{M}^2, \cdots, \mathbf{M}^d]_{i,i}, \quad \mathbf{R}_{i,j} = [\mathbf{I}, \mathbf{M}, \mathbf{M}^2, \cdots, \mathbf{M}^d]_{i,j} \tag{1}$$

$\mathbf{P} \in \mathbb{R}^{N \times d}$ and $\mathbf{R} \in \mathbb{R}^{N \times N \times d}$ are used as the initial node features and edge features in GFSE.

**Pre-training Backbone**. GFSE is built on a GPS architecture (Rampášek et al., 2022) for pre-training, due to its scalability and generalizability. Each GPS layer contains local message passing and global attention modules to capture both neighbor and long-range information. In the $\ell$-th layer, the node encoding $\mathbf{P}$ and relative edge encoding $\mathbf{R}$ are fed into both message passing layers (MPNN) and Biased Attention Module (BiasAttn) parallelly.

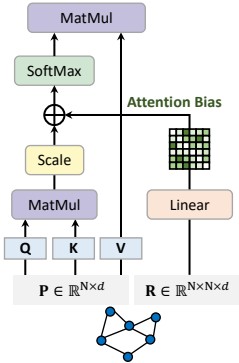

$$\mathbf{P}_M^{\ell+1}, \mathbf{R}^{\ell+1} = \text{MPNN}^\ell\left(\mathbf{P}^\ell, \mathbf{R}^\ell, \mathbf{A}\right), \mathbf{P}_T^{\ell+1} = \text{BiasAttn}^\ell\left(\mathbf{P}^\ell, \mathbf{R}^\ell\right) \quad (2)$$

The node encoding is then updated by $\mathbf{P}^{\ell+1} = \text{MLP}^\ell\left(\mathbf{P}_M^{\ell+1} + \mathbf{P}_T^{\ell+1}\right)$.

**Attention Bias**. The global attention in the original GPS framework does not account for relative edge encoding while leaving them entirely for the message-passing layers. However, incorporating relative edge encoding in global attention is crucial for capturing long-range dependencies, as the receptive field of message-passing layers is inherently constrained by their depth. As shown in Figure 2, to build a theoretically more powerful GPS, we explicitly incorporate relative edge encoding into global attention, where the attention weight between the $i$-th and the $j$-th nodes is computed by $a'_{i,j} = \text{SoftMax}(a_{i,j} + \text{Linear}(\mathbf{R}_{i,j}))$, where $\text{Linear}: \mathbb{R}^d \rightarrow \mathbb{R}$ indicates a linear layer that maps the $d$-dimensional relative encoding to a scalar. $a_{i,j}$ denotes the original attention weight computed by scaled-dot self-attention on the node encoding $\mathbf{P}^\ell$ in each GPS layer.

Figure 2: Biased Attention based on random walk matrix.

### 3.2 SELF-SUPERVISED PRE-TRAINING TASKS

GFSE is pre-trained with four structural tasks, including *reconstruction* and *contrastive learning*. Each task highlights a specific structural aspect, thereby augmenting the model's expressiveness and capability to capture complex graph structures. Let $\mathbf{P}^L \in \mathbb{R}^{N \times d_e}$ represent the output after $L$ GPS layers. We decode $\mathbf{P}^L$ with independent MLP heads for each pre-training task.

**Shortest Path Distance Regression** is an *edge-level reconstruction* task. Shortest Path Distance (SPD) encodes the global proximity and connectivity between nodes, which helps to discern nodes' positions and relations within the entire graph (Li et al., 2020). We pre-compute the shortest path distance via the Dijkstra algorithm (Dijkstra, 1959) to create the label $\text{SPD} \in \mathbb{R}^{N \times N}$. The loss for shortest path distance regression is computed by $\mathcal{L}_{\text{SPD}} = \frac{1}{|E|} \sum_{i,j \in V} \|h_{\text{SPD}}(\mathbf{P}_i^L \| \mathbf{P}_j^L) - \text{SPD}_{i,j}\|^2$, where $\|$ indicates the concatenation operation and $h_{\text{SPD}}$ indicates a task-specific head.

**Motif Counting** is a *node-level reconstruction* task, which allows the model to better identify each node's role in the surrounding subgraphs. We follow previous works (Bouritsas et al., 2022) to count the number of certain motifs surrounding each node. To improve expressiveness, we include a variety of small motifs, called *graphlets* (Pržulj et al., 2004; Pržulj, 2007), with different numbers of vertices, which are beyond usual types like stars, paths, cycles, and cliques. We refer to Appendix B.2 for more technical details. Let $Y_i^m \in \mathbb{Z}^k$ denote the node-level motif label, where $k$ is the number of graphlet types. The loss is formulated as $\mathcal{L}_{\text{MC}} = \frac{1}{|V|} \sum_{i \in V} \|h_{\text{MC}}(\mathbf{P}_i^L) - Y_i^m\|^2$, where $h_{\text{MC}}: \mathbb{R}_e^d \rightarrow \mathbb{R}^k$ is the task-specific head for motif counting.

**Community Detection** is an *edge-level contrastive learning* task that aims to identify densely connected subgraphs, where nodes within a community are more closely linked to each other than to nodes outside the community. Such community structures are ubiquitous in various real-world networks, *e.g.,* social networks, and transportation systems. We employ the Louvain Community Detection Algorithm (Blondel et al., 2008) to extract the community structure from pre-training graphs, which clusters nodes into communities based solely on graph topology without node features. We approach this task in a contrastive learning manner by minimizing the embedding distances between intra-community nodes while maximizing the distance between inter-community nodes by

$$\mathcal{L}_{\text{CD}} = \sum_{i \in V} \sum_{j \in V} Y_{i,j}^c(1 - \text{sim}(i,j)) + (1 - Y_{i,j}^c)\max(0, \epsilon - (1 - \text{sim}(i,j))) \quad (3)$$

where the similarity score $\text{sim}(i,j)$ is calculated by $\text{sim}(i,j) = \frac{\mathbf{z}_i \cdot \mathbf{z}_j}{\|\mathbf{z}_i\| \cdot \|\mathbf{z}_j\|}$ and $\mathbf{z}_i = h_{\text{CD}}(\mathbf{P}_i^L)$ with a head $h_{\text{CD}}$. $\epsilon$ is a margin hyperparameter. $Y_{i,j}^c$ is a binary label that indicates if the $i$-th node and the

$j$-th node are in the same community. Through Eq. 3, GFSE learns to discern community boundaries and distinguish nodes based on local community structures.

**Graph Contrastive Learning** is a *graph-level contrastive learning* task that aims to distinguish graphs from different datasets. The motivation is rooted in the observation that similar structural characteristics in different domains may exhibit distinct meanings. For example, a subgraph representing a protein interaction network in a biological dataset might correspond to a specific functional module, whereas a similar structure in a social network could represent a tightly-knit community or interest group. Therefore, GFSE distinguishes graphs from different datasets, in addition to performing structural pre-training tasks within a single graph. The loss for this task is computed by

$$\mathcal{L}_{\text{GCL}} = -\log \frac{\exp\left(\text{sim}\left(\boldsymbol{z}_{G_i}, \boldsymbol{z}_{G_j}\right)/\tau\right)}{\sum_{k=1}^{K} \mathbb{1}_{[G_k \not\sim G_i]} \exp\left(\text{sim}\left(\boldsymbol{z}_{G_i}, \boldsymbol{z}_{G_k}\right)/\tau\right)}, \tag{4}$$

where $\tau$ is the temperature, $G_i$ and $G_j$ are from the same dataset, $\boldsymbol{z}_{G_i} = \text{GlobalPool}(h_{\text{GCL}}(\mathbf{P}_{G_i}^L))$ is the output of the global pooling applied to the final layer's representation $\mathbf{P}_{G_i}^L$ for the graph $G_i$, $K$ is the number of negative samples, and $\mathbb{1}_{[G_k \not\sim G_i]}$ is an indicator function that determines whether graphs $G_k$ and $G_i$ originate from different datasets.

**Multi-task Loss Weighing**. Each pre-training task targets a different structure aspect, enabling GFSE to capture a comprehensive understanding of graph topology. For instance, the shortest path distance regression task focuses on learning the global connectivity within graphs, while motif counting delves into the occurrence of specific subgraph patterns. This diverse set of tasks covers a wide range of structural properties, from local neighborhoods to global graph characteristics. Since the loss scale and difficulty vary significantly across tasks, we introduce task-specific uncertainty (Kendall et al., 2018), which is learnable during pre-training to unify the scales of all losses. Task-specific uncertainty is used to automatically balance different pre-training losses, *i.e.*, $\mathcal{L}_{\text{SPD}}, \mathcal{L}_{\text{MC}}, \mathcal{L}_{\text{CD}}$, and $\mathcal{L}_{\text{GCL}}$ (see Appendix C.3 for more details). Moreover, the evolution of uncertainty values provides insights into each task's contribution to the overall pre-training process.

### 3.3 COMBINATION WITH DOWNSTREAM FEATURE ENCODER

**Application on Graphs with Vectorized Features**. GFSE can be readily employed to generate expressive PSE for various graph applications. Let $\mathbf{X}^0 \in \mathbb{R}^{N \times d_x}$ denote the initial node features for a given graph with $N$ nodes and $\mathbf{P}^L \in \mathbb{R}^{N \times d_e}$ denote PSE generated by GFSE, where $d_x$ and $d_e$ are dimensions of node features and PSE, respectively. $\mathbf{P}^L$ can then be concatenated with the initial node features $\mathbf{X}^0$ to create a new feature matrix $\mathbf{X}^{new} = [\mathbf{X}^0 \| \mathbf{P}^L] \in \mathbb{R}^{N \times (d_x + d_e)}$, which augments the node features with structural information. This structure-enriched feature $\mathbf{X}^{new}$ can subsequently be fed into downstream graph models, such as graph neural networks or graph transformers, enhancing their performance on various tasks. For large-scale graphs, where computing PSE for the entire graph may be computationally prohibitive, we thereby sample the neighborhood structure around each node and compute the PSE for these localized subgraphs. This process can be efficiently parallelized, enabling scalable and efficient generation of PSE for large graphs.

**Application on Text-attributed Graphs**. Language models are typically employed to process the text-attributed graphs, where GFSE can be seamlessly applied to incorporate structural information. Given the generated $\mathbf{P}^L \in \mathbb{R}^{N \times d_e}$, an MLP is employed to project $\mathbf{P}^L$ into the embedding space of the language model. This projected PSE is then prepended as a soft token to the associated text, effectively incorporating the graph's structural information into the model input. Subsequently, these structure-enriched tokens are fed into downstream large language models (LLMs), enhancing their performance on graph-related tasks. The process, involving training a lightweight MLP and fine-tuning LLM with Parameter-Efficient Fine-Tuning (PEFT) techniques such as LoRA (Hu et al., 2021), makes it scalable and efficient for large-scale text-attributed graph applications.

### 3.4 EXPRESSIVE POWER OF GFSE

We show that GFSE can generate highly expressive PSE by incorporating relative edge encoding into the attention computation in the Graph Transformer backbone. Specifically, we employ the Structural Encoding enhanced Global Weisfeiler-Lehman test (SEG-WL) (Zhu et al., 2023), a generalized WL test that incorporates relative structural encoding into the isomorphism algorithm, to characterize

the expressiveness of GFSE. For an input graph $G(V, E)$ with node set $V$ and edge set $E$, let $f_P : V \to \mathcal{X}$ and $f_R : V \times V \to \mathcal{X}$ indicate the node-level and edge-level structural encoding, respectively. Different from traditional WL test, SEG-WL updates the node labels at the $t$-th iteration by $g_t(v) = \text{hash}(\{\{(g_{t-1}(u), f_R(v, u)) : u \in V\}\})$ and $g_0(v) = \text{hash}(f_P(v))$. SEG-WL can be viewed as a high-level abstraction of the learning paradigm of our pre-training architecture with biased attention (Eq. 2), where relative structural encoding between any two nodes is considered for updating node representations (Zhu et al., 2023). Let RW($d$)-SEG-WL denote the case that $f_P$ and $f_R$ are determined by $\mathbf{P}$ and $\mathbf{R}$ with $d$ dimension, *i.e.,* $f_P(v_i) = \mathbf{P}_i \in \mathbb{R}^d$ and $f_R(v_i, v_j) = \mathbf{R}_{ij} \in \mathbb{R}^d$ for the $i$-th and $j$-th nodes. We have the following propositions.

**Proposition 3.1.** *RW(d)-SEG-WL ($d \geq 3$) is strictly more expressive than 1-WL in testing non-isomorphic graphs.*

**Proposition 3.2.** *There exist pairs of graphs that RW(d)-SEG-WL can distinguish, but 3-WL can not.*

The theoretical proof and empirical verification are given in Appendix D. RW-SEG-WL is able to distinguish all low-order graphs with orders equal to or less than 8 and successfully distinguishes most strongly regular graphs where 3-WL fails to distinguish. RW-SEG-WL stands as an expressivity upper bound of our proposed GFSE. The pre-training tasks are meticulously designed to push GFSE towards achieving the upper bound established by RW-SEG-WL. These pre-training tasks optimize both node-level and edge-level structural encoding, progressively refining the effectiveness of the model in generating expressive encoding.

### 3.5 COMPUTATIONAL COMPLEXITY

The complexity of developing GFSE comprises two parts: pre-computation of self-supervision labels and pre-training. For the pre-computation, we adopt Dijkstra Fibonacci-heap solution (Dijkstra, 1959) to compute the shortest path distance between node pairs, which results in the time complexity of $\mathcal{O}(|E| + |V| \log |V|)$ with node set $V$ and edge set $E$. A brute-force implementation of the subgraph isomorphism counting of fixed size $t$ is $\mathcal{O}(|V|^t)$. We consider the graphlets with at most 5 nodes. One can also choose special graphlet types, *e.g.,* paths, cycles, and triangles, which can be efficiently enumerated (Giscard et al., 2019). Approximating and scalable algorithms can be further used to accelerate this pre-processing step (Fu et al., 2024; Ying et al., 2020; Pashanasangi & Seshadhri, 2020). For pre-training, the complexity is $\mathcal{O}(|V|^2)$ for full attention computation and $\mathcal{O}(d|V||E|)$ for initial encoding computation of $\mathbf{P}$ and $\mathbf{R}$. Notably, the model's PSE generation process requires less than five minutes for all downstream datasets. See runtime evaluations in Appendix F.3.

## 4 EXPERIMENTS

GFSE is pre-trained to recognize complex structural patterns. We first evaluate the pre-training performance in Sec. 4.2 and empirically assess the expressiveness of GFSE on synthetic datasets (Sec. 4.3). We then evaluate GFSE in a wide range of downstream graph learning tasks in Sec. 4.4. Specifically, we conduct experiments with pre-trained models on molecular datasets in Sec. 4.5 and pre-trained LLMs on text-attributed graphs in Sec. 4.6.

### 4.1 PRE-TRAINING SETUP

**Dataset**. We utilize a diverse collection of cross-domain datasets for pretraining, ensuring a broad spectrum of graph structures and scales, including MolPCBA, MolHIV, MNIST, peptides, ogbn-proteins, Pokec, ogbn-arxiv and ogbn-product (Wu et al., 2018; Bhatia et al., 2016; Mikolov et al., 2013; Szklarczyk et al., 2019; Chiang et al., 2019; Takac & Zabovsky, 2012; Dwivedi et al., 2023; 2022). These datasets cover several real-world graph domains, such as social networks, academic networks, *etc*. Table 7 in Appendix C.1 presents the detailed statistics of datasets used for pre-training. For large-scale graphs, we first partition them into sets of subgraphs by the METIS algorithm (Karypis & Kumar, 1997) to handle scalability issues. Training samples from different datasets are mixed and randomly shuffled to form a large-scale pre-training dataset.

**Pre-training Setting**. The pre-training stage is conducted on the standard train/validation/test splits of the pre-training datasets. The dimension of initial encoding $d$ is set as 8. We adopt GIN (Xu et al., 2018) as the message-passing layer in the GPS and adopt 8 GPS layers with 8 heads and 128 hidden

dimensions in each layer. The output dimension is 64 by default. We use Adam as the optimizer with an initial learning rate of 0.001 and the batch size is set as 256. The maximum training epochs is 100. An early stopping strategy is used to mitigate overfitting. The pre-training is implemented on the NVIDIA A40 48GB GPU. We refer to Appendix C.4 for more details.

## 4.2 PRE-TRAINING EVALUATION

GFSE is pre-trained with four self-supervised learning tasks. We iteratively change the message passing layers (*e.g.,* GatedGCN (Bresson & Laurent, 2017), GCN (Kipf & Welling, 2016a) and GIN (Xu et al., 2018)) and replace the biased attention with traditional self-attention in the default GFSE architecture. We evaluate the pre-training performance on the standard test split from the pre-training datasets with different architectures as shown in Figure 3. Accuracy is used to measure community detection and graph contrastive learning tasks, indicating the proportion of node (graph) pairs that are correctly predicted. MSE and MAE are used for shortest path distance and motif counting tasks. See more details in Appendix C.2.

The four self-supervision tasks emphasize different structural aspects, necessitating both local message passing and global attention. Moreover, we observe a consistent performance boost when applying biased attention with explicit relative edge encoding. GFSE is thereby built on GIN and biased attention as the default architecture.

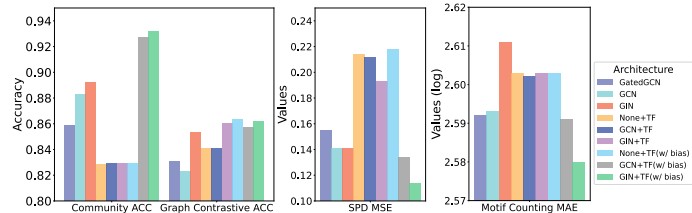

Figure 3: Performance of the pre-trained models with different architectures. MAE for motif counting is shown in the log scale. TF is the abbreviation of transformer.

The pre-trained GFSE is able to generate effective PSE, which can reconstruct other pre-defined PSE, such as LapPE (Lim et al., 2022) and ElstaticPE (Kreuzer et al., 2021). We provide experimental results in Appendix F.1. The ability to reconstruct various types of PSE, without taking them as training objectives explicitly, demonstrates the sufficiency and effectiveness of the chosen tasks in pre-training the model for comprehensive and generalizable graph representations.

## 4.3 EXPRESSIVENESS POWER EVALUATION

We empirically evaluate the structure-awareness of the positional and structural encoding (PSE) generated by GFSE on three benchmarking datasets that require discerning intricate graph topologies. We evaluate the performance boost brought by the PSE generated by GFSE in comparison to two traditional positional encodings, RWSE and LapPE. We test on various downstream graph learning models, including MLP, GIN (Xu et al., 2018), transformer (Vaswani et al., 2017) and GPS (Rampášek et al., 2022). We further compare with the learning-based approach: GPSE (Liu et al., 2023b). See Appendix A for more baseline details. Triangle (Knyazev et al., 2019) poses triangle counting as a 10-way graph-level classification task. Half of the test set are graphs with a similar size to those in the training and validation set (denoted as Triangle-S).

Table 1: Test accuracy (%) enhanced by different positional and structural encoding. The results are averaged over five random seeds. The best results in each dataset are bolded.

| | Triangle-S | Triangle-L | Pattern | Cluster |
|---|---|---|---|---|
| MLP+RWSE | 98.22 | 11.88 | 50.53 | 20.96 |
| MLP+LapPE | 98.60 | 12.62 | 50.53 | 20.96 |
| MLP+GPSE | 52.80 | 17.42 | 55.66 | 20.96 |
| MLP+GFSE | **98.71** | **25.54** | **57.79** | **21.28** |
| GIN | 99.68 | 42.58 | **85.58** | 60.84 |
| GIN+RWSE | 99.70 | 40.78 | 85.34 | 61.30 |
| GIN+LapPE | **99.74** | 42.48 | 85.45 | 61.83 |
| GIN+GPSE | 99.32 | 25.32 | 85.19 | 61.95 |
| GIN+GFSE | 99.72 | **43.84** | 85.58 | **63.49** |
| Transformer (TF) *for Triangle and* GPS *for Others* | | | | |
| TF / GPS | 21.68 | 23.58 | 86.63 | 77.76 |
| TF / GPS+RWSE | 35.96 | 11.38 | 86.68 | 77.72 |
| TF / GPS+LapPE | 35.96 | 12.44 | 86.54 | 77.76 |
| TF / GPS+GPSE | 62.04 | 24.64 | 85.58 | 77.80 |
| TF / GPS+GFSE | **92.82** | **30.15** | **87.98** | **77.86** |

The left are graphs with larger sizes (denoted as Triangle-L), which present greater challenges to the model's expressive power. Pattern and Cluster (Dwivedi et al., 2023) are graph datasets generated with the Stochastic Block Model (SBM) (Abbe, 2018) that require the model to discern graph patterns and local clusters. Both are node-level classification tasks. As shown in Table 1, GFSE generates expressive and robust PSE that consistently improves the base model's performance, whereas other structural encodings exhibit considerable variation across different

datasets or different base models. Notably, the performance boost brought by GFSE is particularly significant for the Transformer on the Triangle dataset, increasing accuracy from 21.68% to 92.82% for Triangle-S. This demonstrates that GFSE has a stronger enhancement effect on models that originally lack structural bias.

## 4.4 DOWNSTREAM EVALUATION

**Dataset and Baseline**. We conduct a comprehensive evaluation of the PSE generated by GFSE on eight real-world graph datasets: MolPCBA, Peptides-func, Peptides-struct, OGB-Arxiv, and MNIST that are in the pre-training dataset, while ZINC (Gómez-Bombarelli et al., 2018), PubMed (Yang et al., 2016) and CIFAR10 (Dwivedi et al., 2023) that are out of the pre-training distribution. We adhere to the experimental setting and hyper-parameters established by previous works (Rampášek et al., 2022) to implement message-passing neural networks, Transformer, and GPS (Rampášek et al., 2022). We augment the initial node features with the PSE generated by GFSE and evaluate the performance on downstream tasks, comparing it against established structural encoding methods including RWSE, LapPE, and GPSE (Liu et al., 2023b).

**Results**. We report the average performance over five random seeds in Table 2 and Table 3 (See standard deviations in Appendix F.2). We observe that the optimal selection of structural encoding typically varies across different datasets and base models. For example, RWSE tends to be more beneficial than LapPE for small molecular graph learning (*e.g.,* MolPCBA and ZINC), while most structural encodings surprisingly degrade the performance on the PubMed dataset. Notably, the performance gains are most pronounced when integrating PSE with Transformer architecture, demonstrating the critical role of structured encodings in compensating for the absence of inherent structural sensitivity in Transformers. The last row shows the average improvement (%) brought by our PSE on base models. The consistent improvements across different settings underscore the robustness and generalizability of GFSE, making it an optimal choice for enhancing the capabilities of graph models, especially in contexts where traditional structural encodings fail to deliver.

Table 2: Performance on MolPCBA, ZINC (subset), Peptides-func and Peptides-struct.

| | MolPCBA AP ↑ | ZINC MAE ↓ | Peptides-func AP ↑ | Peptides-struct MAE ↓ |
|---|---|---|---|---|
| GCN | 0.2424 | 0.3670 | 0.5930 | 0.3496 |
| GCN+LapPE | 0.2417 | 0.2052 | 0.6021 | 0.2688 |
| GCN+RWSE | 0.2438 | 0.1741 | 0.5827 | 0.3270 |
| GCN+GPSE | 0.1958 | **0.1218** | 0.5959 | 0.2710 |
| **GCN+GFSE** | **0.2477** | 0.1237 | **0.6131** | **0.2513** |
| GIN | 0.2703 | 0.5260 | 0.5498 | 0.3547 |
| GIN+LapPE | 0.2701 | 0.2203 | 0.5323 | 0.2650 |
| GIN+RWSE | 0.2781 | 0.1731 | 0.5410 | 0.3282 |
| GIN+GPSE | 0.2765 | 0.2162 | 0.5389 | **0.2581** |
| **GIN+GFSE** | **0.2839** | **0.1689** | **0.5532** | 0.2674 |
| Transformer (TF) | 0.0808 | 0.6943 | 0.4800 | 0.4192 |
| TF+LapPE | 0.1784 | 0.5101 | 0.6307 | 0.2514 |
| TF+RWSE | 0.2083 | 0.2193 | 0.6326 | 0.3344 |
| TF+GPSE | 0.2040 | 0.1883 | 0.6534 | 0.2479 |
| **TF+GFSE** | **0.2376** | **0.1548** | **0.6642** | **0.2436** |
| GPS | 0.2869 | 0.1182 | 0.6535 | 0.2500 |
| GPS+LapPE | **0.2939** | 0.1078 | 0.6494 | 0.2501 |
| GPS+RWSE | 0.2907 | 0.0700 | 0.6603 | 0.2739 |
| GPS+GPSE | 0.2911 | 0.0648 | 0.6688 | **0.2464** |
| **GPS+GFSE** | 0.2916 | **0.0613** | **0.6874** | 0.2474 |
| GFSE Imp.(%) | 32.60 | 76.43 | 2.78 | 42.47 |

Table 3: Test Accuracy (%) on Arxiv, PubMed, MNIST and CIFAR10.

| | Arxiv | PubMed | MNIST | CIFAR10 |
|---|---|---|---|---|
| GateGCN | 71.69 | 76.86 | 97.34 | 67.31 |
| GateGCN+LapPE | 71.95 | 74.83 | 97.10 | 65.08 |
| GateGCN+RWSE | 71.83 | 76.11 | 96.84 | 65.26 |
| GateGCN+GPSE | 72.17 | 71.97 | 96.94 | 65.63 |
| **GateGCN+GFSE** | **72.61** | **78.39** | **97.44** | **68.39** |
| Transformer (TF) | 5.86 | **66.63** | 97.29 | 69.04 |
| TF+LapPE | 5.86 | 66.27 | 96.95 | 69.01 |
| TF+RWSE | 5.86 | 64.43 | 97.81 | 70.70 |
| TF+GPSE | 21.56 | 65.89 | 97.78 | 69.57 |
| **TF+GFSE** | **23.84** | 66.30 | **98.03** | **71.33** |
| GPS | 70.68 | **74.26** | 98.05 | 71.49 |
| GPS+LapPE | 69.51 | 73.68 | 98.16 | 71.87 |
| GPS+RWSE | 72.14 | 72.87 | **98.19** | 71.30 |
| GPS+GPSE | 71.21 | 73.71 | 98.08 | 72.31 |
| **GPS+GFSE** | **72.30** | 74.20 | 98.15 | **74.11** |
| GFSE Imp.(%) | 6.84 | 0.38 | 0.31 | 1.99 |

## 4.5 INTEGRATION WITH PRE-TRAINED SELF-SUPERVISED MODELS ON MOLECULES

**Settings and Methods**. We use the small molecular property prediction datasets namely Tox21, Sider, BBBP, ClinTox, and MUV from the OGB benchmark (Hu et al., 2020a) as a downstream task for GFSE. We evaluate the effectiveness of GFSE under two settings: training from scratch and fine-tuning pre-trained models. In the training from scratch setting, we directly concatenate GFSE's PSE with the raw node features to create new input features. This augmented representation is then fed into a randomly initialized model from the beginning of training. We take GINE (Xu et al., 2018) and GPS (Rampášek et al., 2022) as our backbone. In the fine-tuning setting, we assess GFSE's ability to enhance pre-trained models by concatenating the node encodings obtained from a pre-trained

Table 4: Test ROC-AUC(%) performance on small molecular property prediction datasets. The best results with the same feature encoder for each dataset are bolded.

| | | Tox21 | Sider | BBBP | ClinTox | MUV |
|---|---|---|---|---|---|---|
| Baseline w/o Structural Encoding | SSP | $76.8 \pm 0.8$ | $61.7 \pm 0.8$ | $67.9 \pm 0.9$ | $57.0 \pm 2.8$ | $79.8 \pm 1.6$ |
| | GraphCL | $75.7 \pm 0.5$ | $60.8 \pm 0.7$ | $69.5 \pm 0.5$ | $70.1 \pm 1.9$ | $74.5 \pm 1.3$ |
| | GraphLoG | $75.4 \pm 0.9$ | $61.2 \pm 1.1$ | $72.5 \pm 0.8$ | $76.7 \pm 3.3$ | $76.0 \pm 1.1$ |
| Train From Scratch | GINE | $74.5 \pm 0.4$ | $58.6 \pm 0.1$ | $67.7 \pm 0.7$ | $74.3 \pm 1.5$ | $74.8 \pm 0.6$ |
| | GINE+RWSE | $75.3 \pm 0.2$ | $58.4 \pm 1.8$ | $66.7 \pm 1.4$ | $77.6 \pm 1.4$ | $76.4 \pm 0.8$ |
| | GINE+LapPE | $\mathbf{77.6 \pm 0.8}$ | $57.2 \pm 1.1$ | $65.8 \pm 0.3$ | $75.6 \pm 2.9$ | $77.0 \pm 0.8$ |
| | GINE+GPSE | $74.9 \pm 0.4$ | $60.1 \pm 0.8$ | $66.4 \pm 0.1$ | $78.9 \pm 3.5$ | $75.8 \pm 1.3$ |
| | GINE+GFSE | $75.5 \pm 0.7$ | $\mathbf{60.9 \pm 0.5}$ | $\mathbf{69.1 \pm 1.3}$ | $\mathbf{80.1 \pm 1.5}$ | $\mathbf{77.7 \pm 1.2}$ |
| | GPS | $73.9 \pm 0.2$ | $58.6 \pm 0.4$ | $67.1 \pm 0.3$ | $80.3 \pm 2.4$ | $68.0 \pm 0.6$ |
| | GPS+RWSE | $74.6 \pm 1.3$ | $56.4 \pm 0.6$ | $67.9 \pm 1.0$ | $83.2 \pm 4.6$ | $69.7 \pm 0.6$ |
| | GPS+LapPE | $74.8 \pm 1.1$ | $60.5 \pm 0.6$ | $67.9 \pm 0.6$ | $78.9 \pm 1.4$ | $70.1 \pm 2.2$ |
| | GPS+GPSE | $75.1 \pm 0.7$ | $56.6 \pm 1.7$ | $67.8 \pm 0.7$ | $73.8 \pm 0.7$ | $68.3 \pm 0.1$ |
| | GPS+GFSE | $\mathbf{76.3 \pm 1.4}$ | $\mathbf{61.8 \pm 0.5}$ | $\mathbf{68.0 \pm 0.5}$ | $\mathbf{83.6 \pm 3.8}$ | $\mathbf{73.6 \pm 0.5}$ |
| Fine-tune Pre-trained Models | GraphMAE | $75.4 \pm 0.4$ | $59.8 \pm 0.5$ | $69.5 \pm 1.6$ | $\mathbf{77.4 \pm 2.9}$ | $76.3 \pm 2.4$ |
| | GraphMAE+RWSE | $\mathbf{76.3 \pm 0.5}$ | $60.5 \pm 0.8$ | $66.4 \pm 3.7$ | $76.7 \pm 5.3$ | $77.7 \pm 1.5$ |
| | GraphMAE+GFSE | $75.9 \pm 0.9$ | $\mathbf{62.1 \pm 0.8}$ | $\mathbf{70.5 \pm 1.4}$ | $77.2 \pm 5.2$ | $\mathbf{78.1 \pm 1.3}$ |
| | MoleBERT | $76.8 \pm 0.5$ | $62.8 \pm 1.1$ | $\mathbf{71.9 \pm 1.6}$ | $78.9 \pm 3.0$ | $78.6 \pm 1.8$ |
| | MoleBERT+RWSE | $77.8 \pm 0.7$ | $\mathbf{63.1 \pm 0.6}$ | $66.5 \pm 2.1$ | $73.9 \pm 3.2$ | $80.4 \pm 1.3$ |
| | MoleBERT+GFSE | $\mathbf{78.0 \pm 0.4}$ | $\mathbf{63.1 \pm 0.7}$ | $68.9 \pm 2.1$ | $78.1 \pm 2.1$ | $\mathbf{80.5 \pm 2.0}$ |

model with the PSE generated by GFSE. The concatenated features are then fed into the final read-out layers for prediction. During fine-tuning, the parameters of both the pre-trained model and the read-out layers are continuously updated. We select the pre-trained models, GraphMAE (Hou et al., 2022) and MoleBERT (Xia et al., 2022) as the backbones and compare with other baselines without structural encoding, namely SSP (Hu et al., 2019), GraphLoG (Xu et al., 2021b), GraphCL (You et al., 2020). Refer to Appendix E.1 for more implementation details.

**Results**. Experimental results are shown in Table. 4. For training the models from scratch, on both GINE and GPS, PSE consistently improves model performance, achieving better results than all the other structural feature augmentation methods across all datasets. As to fine-tuning, our PSE significantly boosts the performance of MoleBERT on three out of five datasets and achieves state-of-the-art performance on Tox21, Sider and MUV datasets. In the case of GraphMAE, PSE achieves better performance than RWSE in four out of five datasets, and also significantly enhances the performance of the backbone (GraphMAE) in four out of five datasets.

## 4.6 INTEGRATION WITH LARGE LANGUAGE MODELS

**Settings and Methods**. We perform experiments on e-commerce networks from Amazon (He & McAuley, 2016; McAuley et al., 2015), which are text-attributed graphs with detailed descriptions for each node (*i.e.,* product item). Edges indicate *co-viewed* or *co-purchased* relations between two nodes. The dataset statistics of three selected categories can be found in Table 8. We employ a lightweight MLP to align the PSE generated by GFSE with the language model's embedding space, which ensures seamless integration of structural information into the language model. We concatenate the textual description of a central node with those of its one-hop neighbors and prepend the PSE as a soft token, followed by a special graph token at the end. This combined sequence is then encoded by LLaMA2 (Touvron et al., 2023). The hidden embedding of the special graph token is used as the representation for the central node. Following the previous setting (Zhu et al., 2024), we compute the cosine similarity between the representations of node pairs as the edge likelihood. We train the MLP and fine-tune the language model with LoRA (Hu et al., 2021) using a contrastive loss (Hadsell et al., 2006). More evaluation details can be found in Appendix E.2.

**Results**. Hit@1 and Mean Reciprocal Rank (MRR) results are reported in Table 5. We select InstructGLM (Ye et al., 2023) as a baseline, which has been fine-tuned on graph domains without structural information infusion. Additionally, we include comparisons against GraphSAGE (Hamilton et al., 2017), which was trained from

Table 5: Comparison with general-domain baselines

| | Cloth | | Home | | Sport | |
|---|---|---|---|---|---|---|
| | Hit@1 | MRR | Hit@1 | MRR | Hit@1 | MRR |
| InstructGLM | 76.23 | 82.60 | 79.82 | 85.93 | 62.50 | 73.25 |
| Finetuned LLaMA | 74.73 | 82.87 | 78.93 | 86.07 | 62.52 | 75.77 |
| + GraphSAGE | 76.22 | 84.16 | 73.74 | 81.66 | 62.26 | 75.36 |
| + GFSE (Ours) | **76.84** | **84.68** | **79.85** | **86.77** | **64.79** | **76.24** |

scratch as a PSE encoder with LLaMA finetuning together. The term *Finetuned LLaMA* refers to the LLaMA model fine-tuned without incorporating any PSE. As seen from the table, GFSE, which is pre-trained on cross-domain graph data as a structural encoder, consistently outperforms other methods across all datasets. In particular, GFSE provides an average boost of $2.01\%$ in performance over InstructGLM and a $3.53\%$ improvement over GraphSAGE encoder across the three datasets. These gains demonstrate the effectiveness of GFSE in graph-based language modeling tasks. See more discussions in Appendix F.2.

### 4.7 ABLATION STUDIES

We analyze the sensitivity and effect of each pre-training task and model architecture in terms of the performance boost of the generated PSE in downstream tasks. The results are illustrated in Table 6. Firstly, we iteratively remove one of the four pre-training tasks and follow the same setting to pre-train and evaluate GFSE. We observe that the removal of each task results in a discernible reduction in downstream task performance. Notably, the shortest path distance task is particularly critical for the ZINC and CIFAR10 datasets, while local community detection plays a significant role in enhancing performance on academic datasets.

Table 6: Ablation studies on the pre-training tasks and model architecture of GFSE. Best results are shown in bold.

| | ZINC MAE ↓ | CIFAR10 ACC ↑ | ogbn-arxiv ACC ↑ |
|---|---|---|---|
| **GPS w/o PE** | 0.1182 | 71.49 | 70.68 |
| **Augment by GFSE** | **0.0613** | **74.11** | 72.30 |
| *Pre-training Tasks* | | | |
| *w/o* Community Detection | 0.0637 | 72.38 | 70.34 |
| *w/o* Motif Counting | 0.0731 | 73.02 | 71.27 |
| *w/o* Shortest Path Distance Regression | 0.1074 | 71.58 | 72.06 |
| *w/o* Graph Contrastive Learning | 0.0856 | 73.02 | 72.13 |
| *Model Architecture* | | | |
| GIN+Traditional Attention | 0.0872 | 73.13 | 71.85 |
| Biased Attention Only | 0.1137 | 70.97 | 71.33 |
| GIN Only | 0.0640 | 72.31 | **72.34** |

We further conduct ablation studies on the main components of GFSE. Specifically, we use traditional attention to replace biased attention as a baseline and remove GIN or attention modules respectively. We notice that all the above pre-training architecture variants lead to performance degradation. The hybrid approach outperforms both attention-only and GIN-only setups, suggesting that integrating sophisticated attention mechanisms can compensate for the absence of global information in local message-passing layers. Figure 4 illustrates the trajectory of task uncertainty ($\sigma^2$) across different pre-training tasks *w.r.t.* pre-training epochs. Higher values of $\sigma^2$ reduce the respective task's contribution to the overall training loss. We observe that all tasks show a sharp decline in

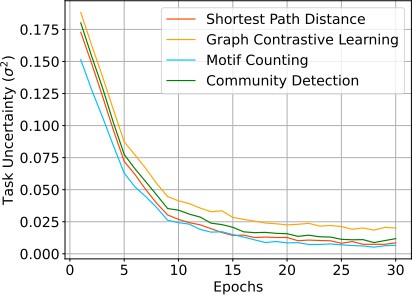

Figure 4: Learning task uncertainty ($\sigma^2$) *w.r.t.* pre-training epochs

uncertainty during the pre-training process. Notably, motif counting maintains a lower uncertainty throughout the training process compared to other tasks, suggesting that it might be inherently more straightforward for the model to optimize or more integral to the model's overall learning structure.

## 5 CONCLUSION

GFSE represents a significant advancement in cross-domain graph foundation models, leveraging multiple self-supervised learning objectives to capture comprehensive structural information from diverse graph domains. By integrating relative positional encoding within a Graph Transformer, GFSE provides a robust framework for generating expressive positional and structural encodings. Extensive experiments on synthetic and real-world datasets validate GFSE's ability to enhance the performance of various graph feature encoders, broadening its applicability across numerous graph-related tasks. Building upon the promising results of GFSE, one potential direction is to explore the impact of pre-training dataset diversity on the model's ability to capture multi-level topological features. Investigating techniques to curate more representative and varied pre-training datasets could further enhance GFSE's generalization capabilities across different graph domains.

**Reproducibility Statement**. The code is available at `https://anonymous.4open.science/r/GFSE-E8C0`. Detailed descriptions of the datasets used in our experiments, along with the specific data processing steps, can be found in Appendix B and Appendix C.1. Pre-training setting can be found in Appendix C.4.

**Ethics Statement**. Our work does not raise significant ethical concerns. The datasets used are publicly available, and we comply with all privacy and legal standards in their use. No human subjects were involved in this study, and there are no potential conflicts of interest or sponsorship issues.

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

# A   POSITIONAL AND STRUCTURAL ENCODING

**Random Walk Structural Encoding**. Random walk structural encoding (RWSE) (Dwivedi et al., 2021; Rampášek et al., 2022) uses landing probabilities derived from random walks of varying lengths, starting from each node in the graph to capture both structural and positional relationships among nodes. Formally, let $G(V, E)$ represent an input graph, where $V$ and $E$ denote the set of $N$ nodes and edges, respectively. $\mathbf{A} \in \mathbb{R}^{N \times N}$ indicates the adjacency matrix and $\mathbf{D}$ is the degree matrix. Random Walk matrix is defined as $\mathbf{M} = \mathbf{D}^{-1}\mathbf{A}$, where $\mathbf{M}_{i,j}$ indicates the transition probability from the $i$-th node to the $j$-th node. The RWSE with $d$ steps for the $i$-th node is defined as

$$\text{RWSE}_i = [\mathbf{I}, \mathbf{M}, \mathbf{M}^2, \cdots, \mathbf{M}^d]_{i,i}, \tag{5}$$

**Laplacian Positional Encoding**. Laplacian positional encoding (LapPE) (Kreuzer et al., 2021; Dwivedi et al., 2023) method emerges as a significant advancement for enriching node representations with spectral information. LapPE utilizes the eigenvectors of the graph Laplacian matrix to encode the relative positions of nodes within a graph. These eigenvectors reflect a local coordinate system that captures meaningful structural information, while also preserving the global topological properties of the original graph. The Laplacian matrix $\mathbf{L} = \mathbf{D} - \mathbf{A}$ has the full eigendecomposition as $\mathbf{L} = \mathbf{U}\boldsymbol{\Lambda}\mathbf{U}^{\top}$. We use the $k$ smallest non-trivial eigenvectors of the Laplacian matrix to create the LapPE. The LapPE with $k$ eigenvectors for the $i$-th node is expressed as:

$$\text{LapPE}_i = [\boldsymbol{u}_{1,i}, \boldsymbol{u}_{2,i}, \cdots, \boldsymbol{u}_{k,i}] \in \mathbb{R}^k, \tag{6}$$

where $\boldsymbol{u}_t$ denotes the $t$-th smallest non-trivial eigenvectors and $k$ is the number of eigenvectors used. Laplacian PE is particularly useful in situations where nodes are inherently anonymous and lack unique features. However, the Laplacian encoding faces challenges from the arbitrary sign of normalized eigenvectors, introducing $2^k$ possible configurations for $k$ eigenvectors. To manage this complexity during training, eigenvectors are randomly sampled from these possibilities. Alternatively, resolving the sign ambiguity by taking the absolute values of eigenvectors simplifies the model but can significantly reduce the expressiveness of the positional features.

**Pre-trained Positional and Structural Encoder**.   Graph Positional and Structural Encoder (GPSE) (Liu et al., 2023b) is a graph encoder pre-trained on molecule datasets by reconstructing traditional positional encoding, such as LapPE, RWSE, CycleSE, *etc*. The model takes as input graph adjacency matrix and randomly generates node features by $\mathbf{X} \sim \mathcal{N}(\mathbf{0}, \mathbf{I})$ to improve the expressiveness. The pre-training architecture is a deep MPNN with 20 layers and residual connection and gating mechanism. Moreover, GPSE utilizes a virtual node technique in each graph to enable global message passing. However, GPSE suffers from poor generalizability across other domains, due to its pretraining setting and randomized node features.

# B   DATASET PRE-PROCESSING

## B.1   SHORTEST PATH DISTANCE REGRESSION

Shortest Path Distance (SPD) Regression is an edge-level reconstruction task. SPD encodes the global proximity and connectivity between nodes, which helps to discern nodes' positions and relations within the entire graph (Li et al., 2020). For data preprocessing, we pre-calculate the $N \times N$ SPD matrix of a given graph before the pretraining phase and save the matrix with the graph data for fast retrieval. We utilize the Dijkstra algorithm (Dijkstra, 1959) to compute the shortest path distances between node pairs, which serve as SPD labels. During pretraining, we randomly select node pairs and fetch their SPD as labels to perform the edge-level reconstruction task. The loss for shortest path distance regression is computed as:

$$\mathcal{L}_{\text{SPD}} = \frac{1}{|E|} \sum_{i,j \in V} \|h_{\text{SPD}}(\mathbf{P}_i^L \| \mathbf{P}_j^L) - \text{SPD}_{i,j}\|^2$$

where $\|$ denotes the concatenation operation and $h_{\text{SPD}}$ indicates the task-specific head.

## B.2 MOTIF COUNTING

Motif Counting is formulated as a node-level reconstruction task. We explicitly use subgraph isomorphism counting as a form of self-supervision during pretraining, allowing the model to better leverage structural information and identify each node's role in the surrounding subgraphs (Bouritsas et al., 2022). We follow previous works on subgraph isomorphism to count the number of certain motifs surrounding each node. A key concept in our preprocessing step is "Automorphism orbits," as introduced by Pržulj (Pržulj, 2007). This concept helps in identifying unique roles of nodes within the counted motifs. The detailed counting method is as follows:

1. **Define the subgraph structures to be counted:** Enumerate all graphlets with the number of nodes less than or equal to 4. This limitation is due to the exponential growth in the number of graphlets with increasing nodes, which becomes computationally infeasible.

2. **Assign indexed orbits:** For each subgraph (or motif), assign each vertex a uniquely indexed orbit to facilitate accurate counting.

3. **Count specified subgraphs:** Utilize the `subgraph_isomorphism` function from the Python package `graph-tool` to count specified subgraphs throughout the entire graph. This count is used to determine orbits for each node in the entire graph, forming an orbit degree vector.

4. **Save and prepare node-level labels:** Save these vectors with the data as node-level labels, which are then prepared for our node-level reconstruction task.

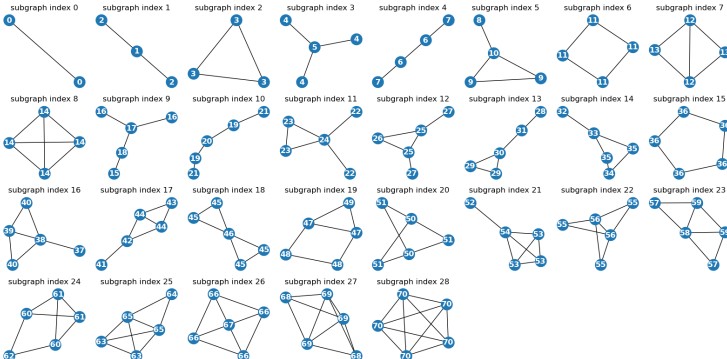

Figure 5: Illustration of various subgraphs (graphlets) used in the motif counting. Each subgraph is indexed and labeled for reference.

## C EXPERIMENTS

### C.1 DATASET

**MolPCBA** dataset consists of 437,929 molecular graphs, each representing a compound from the PubChem BioAssay database. This dataset is used for multi-label binary classification tasks across 128 targets, focusing on predicting the bioactivity of compounds against various protein targets. The primary evaluation metric for this dataset is Average Precision (AP).

**MolHIV** contains 41,127 molecular graphs derived from the MoleculeNet benchmark. Each graph represents a molecule, with nodes as atoms and edges as chemical bonds. The task is a binary classification to predict the ability of molecules to inhibit HIV replication, with AUROC as the evaluation metric.

**MNIST** dataset includes 70,000 images converted into graphs. Each image represents a handwritten digit, with nodes representing pixels and edges representing pixel adjacency. The task is a 10-way classification to identify the digit in the image, evaluated using accuracy (ACC).

Table 7: Pretraining Dataset Information. *class.* represents classification task and *reg.* represents regression task.

| Dataset | Num. graphs | Num. nodes | Num. edges | Pred. level | Pred. task | Num. tasks | Metric |
|---|---|---|---|---|---|---|---|
| MolPCBA | 437,929 | 25.97 | 28.11 | graph | class. (binary) | 128 | AP |
| MolHIV | 41,127 | 25.51 | 27.46 | graph | class. (binary) | 1 | AUROC |
| MNIST | 70,000 | 70.57 | 281.65 | graph | class. (10-way) | 1 | ACC |
| Peptides-func | 15,535 | 150.94 | 153.65 | graph | class. (binary) | 10 | AP |
| Peptides-struct | 15,535 | 150.94 | 153.65 | graph | reg. | 11 | MAE |
| ogbn-proteins | 1 | 132,534 | 39,561,252 | node | class. (binary) | 112 | AUROC |
| Pokec | 1 | 1,632,803 | 30,622,564 | node | class. (binary) | 1 | ACC |
| ogbn-arxiv | 1 | 169,343 | 1,166,243 | node | class. (40-way) | 1 | ACC |
| ogbn-products | 1 | 2,449,029 | 61,859,140 | node | class. (47-way) | 1 | ACC |
| ZINC | 249,456 | 23.2 | 49.8 | graph | reg. | 1 | MAE |
| PubMed | 19,717 | 88,648 | 500 | node | class. (3-way) | 1 | ACC |
| CIFAR10 | 60,000 | 117.6 | 941.2 | graph | class. (10-way) | 1 | ACC |

**Peptides-func** comprises 15,535 graphs, each representing a peptide. Nodes represent amino acids, and edges represent peptide bonds. The task involves binary classification to predict the functional properties of the peptides, with Average Precision (AP) as the evaluation metric.

**Peptides-struct** also contains 15,535 peptide graphs but focuses on regression tasks to predict structural properties of peptides, such as bond angles and distances. The evaluation metric is Mean Absolute Error (MAE).

**ogbn-proteins** dataset is a large-scale graph with 132,534 nodes and 39,561,252 edges, representing protein-protein interaction networks. Each node is a protein, and edges represent interactions. The task is binary classification at the node level to predict protein functions, evaluated using AUROC.

**Pokec** is a social network dataset from the Pokec online social network in Slovakia. It includes one large graph with 1,632,803 nodes (users) and 30,622,564 edges (friendships). The task is binary classification to predict user attributes, such as gender, with accuracy (ACC) as the metric.

**ogbn-arxiv** dataset consists of a single large graph with 169,343 nodes and 1,166,243 edges, representing the citation network of arXiv papers. Each node is a paper, and edges represent citation links. The task is 40-way classification to predict the primary subject area of each paper, evaluated using accuracy (ACC).

**ogbn-products** dataset includes a large graph with 2,449,029 nodes and 61,859,140 edges, representing an Amazon product co-purchasing network. Nodes represent products, and edges represent co-purchasing relationships. The task is 47-way classification to predict the product category, with accuracy (ACC) as the metric.

**ZINC** contains 249,456 molecular graphs, where each graph represents a molecule from the ZINC database. The task is regression to predict molecular properties like solubility, with Mean Absolute Error (MAE) as the evaluation metric.

**PubMed** dataset includes a citation network of 19,717 scientific publications. Nodes represent papers, and edges represent citations. The task is 3-way classification to predict the subject areas of the papers, evaluated using accuracy (ACC).

**CIFAR10** dataset has 60,000 images transformed into graphs, where each image represents a colored object. Nodes represent pixels, and edges represent pixel adjacency. The task is a 10-way classification to identify the object in the image, evaluated using accuracy (ACC).

### C.2 PRE-TRAINING METRIC

We use an accuracy metric to measure Community Detection and Graph Contrastive Learning and use mean squared error (MSE) to measure the performance of Shortest Path Distance regression and Motif Counting.

Table 8: Dataset statistics of three categories from Amazon e-commerce networks

| | # nodes | # edges | avg. degree | avg. # tokens |
|---|---|---|---|---|
| **Clothing** | 469,274 | 2,578,746 | 10.99 | 117.83 |
| **Home** | 453,121 | 3,732,948 | 16.48 | 133.94 |
| **Sports** | 293,712 | 2,390,076 | 16.27 | 125.08 |

Table 9: Test performance of the pre-trained model with different architectures on four pre-training tasks.

| Architecture | Community Detection Accuracy | Graph Contrastive Elanring Accuracy | Shortest Path Distance MSE | Motif Counting MAE (log) |
|---|---|---|---|---|
| **GatedGCN** | 0.859 | 0.831 | 0.155 | 2.592 |
| **GCN** | 0.883 | 0.823 | 0.141 | 2.593 |
| **GIN** | 0.892 | 0.853 | 0.141 | 2.611 |
| **None+TF** | 0.828 | 0.841 | 0.214 | 2.603 |
| **GCN+TF** | 0.829 | 0.841 | 0.212 | 2.602 |
| **GIN+TF** | 0.829 | 0.860 | 0.193 | 2.603 |
| **None+TF(w/ bias)** | 0.829 | 0.863 | 0.218 | 2.603 |
| **GCN+TF(w/ bias)** | 0.927 | 0.857 | 0.134 | 2.591 |
| **GIN+TF(w/ bias)** | **0.932** | **0.862** | **0.114** | **2.580** |

- For the community detection task, we set $\epsilon$ as 1 in Eq. 3. The predicted label between the $i$-th node and the $j$-th node $\hat{Y}_{i,j}^c$ is 1 if $\text{sim}(i, j) \geq 0.5$ and 0 otherwise. Accuracy is calculated by comparing the predicted label $\hat{Y}_{i,j}^c$ with the ground truth labels $Y_{i,j}^c$ and is defined as the proportion of correctly predicted labels out of all possible node pairs:

$$\text{Accuracy(CD)} = \frac{\sum_{(i,j) \in V} \mathbb{1}(\hat{Y}_{i,j}^c = Y_{i,j}^c)}{|V|(|V| - 1)/2}, \tag{7}$$

where $\mathbb{1}(\cdot)$ is an indicator function and $|V|(|V| - 1)/2$ is the total number of unique node pairs in the graph. This metric effectively measures how well the model can identify community structures by correctly classifying node pairs as being in the same community or in different communities.

- For the graph contrastive learning task, we evaluate pre-training performance using the accuracy metric, which measures the model's ability to correctly classify graphs as originating from the same or different datasets. The accuracy is computed by:

$$\text{Accuracy(GCL)} = \frac{\sum_{i,j} \mathbb{1}(\hat{Y}_{G_i,G_j} = Y_{G_i,G_j})}{N}, \tag{8}$$

where $\hat{Y}_{G_i,G_j}$ is the predicted label indicating whether graph $G_i$ and $G_j$ are from the same dataset and $Y_{G_i,G_j}$ is the ground truth label. $\hat{Y}_{G_i,G_j}$ is 1 if $\text{sim}(\boldsymbol{z}_{G_i}, \boldsymbol{z}_{G_j}) \geq 0$ and 0 otherwise. $N$ is the total number of evaluated graph pairs.

- For shortest path distance regression, the mean squared error (MSE) is used as a metric, which is defined as:

$$\text{MSE(SPD)} = \frac{1}{|E|} \sum_{(i,j) \in E} (h_{\text{SPD}}(\mathbf{P}_i^L \| \mathbf{P}_j^L) - \text{SPD}_{i,j})^2. \tag{9}$$

The ground truth SPD is normalized by the graph diameter to ensure scale consistency and training stability.

- For the motif counting task, the mean absolute error is used as a metric, which is defined as:

$$\text{MAE(MC)} = \frac{1}{|V|} \sum_{i \in V} \|h_{\text{MC}}(\mathbf{P}_i^L) - Y_i^m\|_1, \tag{10}$$

where $Y_i^m$ is the pre-computed label for the $i$-th node.

## C.3 UNCERTAINTY-BASED LOSS WEIGHING

The scale of the loss of different tasks can be different, causing the overall loss to be dominated by a certain task, and ultimately the loss of the other tasks cannot affect the learning process of the

network-sharing layers. We use the uncertainty-based loss-weighing method (Kendall et al., 2018) to automatically balance the four pre-training tasks and unify the different scales. Moreover, the uncertainty value reflects the contribution of each task towards the overall pre-training process. A higher uncertainty value indicates a lower contribution (Kendall et al., 2018). Let $\sigma_\tau$ and $\mathcal{L}_\tau$ represent the task-specific uncertainty value for the task $\tau$. The overall pre-training loss is computed by:

$$
\begin{aligned}
\mathcal{L} = &\frac{1}{\sigma_{\text{SPD}}^2}\mathcal{L}_{\text{SPD}} + \frac{1}{\sigma_{\text{MC}}^2}\mathcal{L}_{\text{MC}} + \frac{1}{\sigma_{\text{CD}}^2}\mathcal{L}_{\text{CD}} + \frac{1}{\sigma_{\text{GCL}}^2}\mathcal{L}_{\text{GCL}} \\
&+ \log \sigma_{\text{SPD}} + \log \sigma_{\text{MC}} + \log \sigma_{\text{CD}} + \log \sigma_{\text{GCL}}.
\end{aligned}
\tag{11}
$$

### C.4 PRE-TRAINING SETTING

The pre-training stage is conducted on the standard train/validation/test splits of the pre-training datasets. The dimension of initial encoding $d$ is set as 8. We try GatedGCN (Bresson & Laurent, 2017), GCN (Kipf & Welling, 2016a) and GIN (Xu et al., 2018) as the message-passing layers in the GPS. The number of GPS layers is tuned in the range of [4, 16] and the number of heads is tuned within $\{4, 8, 16\}$. The hidden dimension is tuned $\{32, 64, 128, 256\}$. The output PSE dimension is in $\{32, 64\}$. The temperature $\tau$ in Eq. 4 is set as $0.1$ and the margin $\epsilon$ in Eq. 3 is 0. We use Adam as the optimizer with an initial learning rate of $0.001$ and the batch size is set as 256. The maximum training epochs is 100. An early stopping strategy is used to mitigate overfitting. The pre-training and downstream evaluation are implemented on the NVIDIA A40 48GB GPU. Experiments on the molecule dataset run on a server with one AMD EPYC 7763 64-Core processor and a NVIDIA RTX 6000 GPU card. The code is available at the following anonymous link: `https://anonymous.4open.science/r/GFSE-E8C0`.

## D EXPRESSIVENESS

### D.1 THEORETICAL PROOF

For an input graph $G(V, E)$ with node set $V$ and edge set $E$, let $f_P : V \to \mathcal{X}$ and $f_R : V \times V \to \mathcal{X}$ indicate the node-level and edge-level structural encoding, respectively. SEG-WL updates the node labels at the $t$-th iteration by $g_t(v) = \text{hash}(\{\{(g_{t-1}(u), f_R(v, u)) : u \in V\}\})$ and $g_0(v) = \text{hash}(f_P(v))$.

**Proposition D.1.** *RW(d)-SEG-WL ($d \geq 3$) is strictly more expressive than WL in testing non-isomorphic graphs.*

**Proposition D.2.** *There exist pairs of graphs that RW(d)-SEG-WL can distinguish, but 3-WL can not.*

*Proof.* We first introduce Neighbor-SEG-WL, which is the SEG-WL test when $f_P$ is an identity encoding and $f_R(u, v)$ equals 1 if $(u, v) \in E$ and 2 otherwise. Previous works have proved the following Proposition (Zhu et al., 2023).

**Proposition D.3.** *Two non-isomorphic graphs can be distinguished by WL if and only if they are distinguishable by Neighbor-SEG-WL.*

Therefore, Neighbor-SEG-WL is a specific example of SEG-WL test that has equivalent expressiveness to the 1-WL test. We then prove that RW-SEG-WL is strictly more expressive than Neighbor-SEG-WL. Let $d_{\text{neg}}(u, v)$ indicate the edge-level encoding $f_R$ in Neighbor-SEG-WL. Note $d_{\text{neg}}(v_i, v_j) = 2$ if and only if $A_{ij} = 0$. Recall that $f_R(\cdot, \cdot)$ in RW-SEG-WL satisfies $f_R(v_i, v_j) = \mathbf{R}_{ij} \in \mathbb{R}^d$ with $\mathbf{R} = [\mathbf{I}, \mathbf{M}, \cdots, \mathbf{M}^d]$ where $\mathbf{M} = \mathbf{D}^{-1}\mathbf{A}$. Therefore, $f_R$ in RW-SEG-WL strictly contains the information of $d_{\text{neg}}$. Therefore, if two non-isomorphic graphs can be distinguished by WL, they can be distinguished by RW-SEG-WL. Proposition D.1 is proved.

To prove Proposition D.2, we provide an example in Figure 6 which shows the Shrikhande graph and the Rook's $4 \times 4$ graph, a pair of strongly regular graphs SRG(16,6,2,2). It is proved that they cannot be distinguished by 3-WL (Arvind et al., 2020). We empirically verified that RW(d)-SEG-WL with $d > 4$ can distinguish these two graphs. □

### D.2 SYNTHETIC GRAPH ISOMORPHISM TESTS

To evaluate the expressive power of RW-SEG-WL, we perform synthetic graph isomorphism tests on low-order graphs and strongly regular graphs. We consider low-order graphs with up to 8 nodes. Strongly regular graph SRG$(n, k, \lambda, \mu)$ means graphs with $n$ nodes, where each node has $k$ neighbors. Each adjacent pair of nodes has the same number $\lambda$ of neighbors in common and each non-adjacent node pair has $\mu$ neighbors in common. Strongly regular graphs are known to be challenging cases for graph isomorphism test algorithms due to their highly symmetric structure. We compare with 1-WL and SPD-SEG-WL, where $f_P$ is an identity encoding and $f_R$ is defined as the shortest path distance between

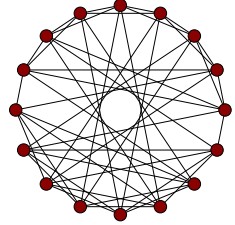 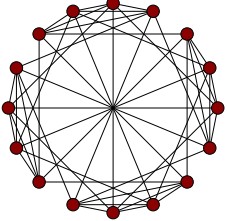

(a) The Shrikhande graph      (b) The Rook's 4×4 graph

Figure 6: RW($d$)-SEG-WL can distinguish the Shrikhande graph and the Rook's $4 \times 4$ graph when $d > 4$ while 3-WL fails

two nodes. Note SPE-SEG-WL can be viewed as an expressivity upper bound of Graphormer (Ying et al., 2021). The results are shown in Table 10. We observe that RW-SEG-WL can distinguish significantly more non-isomorphic graphs than 1-WL and SPD-SEG-WL. Specifically, with $d$ equals 8, *i.e.,* when considering random walk with 8 steps, RW-SEG-WL successfully distinguish all low-order graphs and strongly regular graphs. When setting $d = 4$, there are 16 pairs of strongly regular graphs that cannot be distinguished by RW-SEG-WL. Therefore, it is natural to develop a graph transformer equipped with a relative random-walk encoding that can accurately capture important graph structures and demonstrate strong expressive power.

Table 10: Results of synthetic graph isomorphism tests

| | Low-Order Graphs (Parameter:$n$) | | | | Strongly Regular Graphs (Parameter:$(n, k, \lambda, \mu)$) | | | | | |
|---|---|---|---|---|---|---|---|---|---|---|
| Parameter | 5 | 6 | 7 | 8 | (25,12,5,6) | (26,10,3,4) | (29,14,6,7) | (36,14,4,6) | (40,12,2,4) | 45,12,3,3 |
| # graphs | 21 | 112 | 853 | 11117 | 15 | 10 | 41 | 180 | 28 | 78 |
| # graph pairs | 210 | 6216 | 363378 | 61788286 | 105 | 45 | 820 | 16110 | 378 | 3003 |
| **number of undistinguishable graph pairs** | | | | | | | | | | |
| WL | 0 | 3 | 17 | 312 | 105 | 45 | 820 | 16110 | 378 | 3003 |
| SPD-SEG-WL | 0 | 2 | 12 | 186 | 105 | 45 | 820 | 16110 | 378 | 3003 |
| RW-SEG-WL($d = 8$) | 0 | 0 | 0 | 0 | 0 | 0 | 0 | 0 | 0 | 0 |

# E  EVALUATION DETAILS

## E.1  INTEGRATION WITH PRE-TRAINED MODELS ON MOLECULES

All models are fine-tuned, trained or tested using five different seeds from 42 to 46, with the results averaged. Additionally, for the results of our generated PSE, we select three different seeds to obtain three GFSE checkpoints. Each GFSE is used to run the downstream task five times with the aforementioned seeds ($42 - 46$), and all results are averaged.

For the training from scratch setting, we adopt and modify the code base from GPS (Rampášek et al., 2022) [1]. RWSE and LapPE are of dimension 32 for molecule benchmark in Table 4 across all the datasets. For other downstream graph tasks in Table 2 and Table 3, we follow exactly as the hyper-parameters established in GPS (Rampášek et al., 2022). Given a graph $G(V, E)$, we directly concatenate the PSE and the raw node feature as the new input feature, then send them into the very beginning of a model with randomly initialized parameters, which is as follows:

$$X' = \text{concat}(X, \text{PSE}) \tag{12}$$

$$\hat{y} = \text{MLP}[\text{pooling}[\text{GraphModel}(X')]], \tag{13}$$

where $X$ denotes the raw node feature of the input graph, $X'$ is the input feature augmented with structural information, $\text{GraphModel}$ denotes our backbones GNN or GPS, and the read-out layer consists of pooling and MLPs to obtain the final prediction.

---

[1] https://github.com/rampasek/GraphGPS

For the fine-tuning setting in the molecule benchmark, we concatenate the node encodings obtained from a pre-trained model with the extra structural features. Then we send the concatenated features into the final read-out layers for the final prediction. Note that during the fine-tuning process, the parameters of the entire model (both the pre-trained model and the read-out layer) are continuously updated.

$$X' = \text{GNN}(X) \tag{14}$$

$$\hat{y} = \text{MLP}[\text{pooling}[\text{concact}(X', \text{PSE})]], \tag{15}$$

where $X'$ denotes the latent node features output from the pre-trained models, PSE denotes the extra structural feature generated by GFSE, the read-out layer consists of pooling and MLPs, where the hyper-parameters follow exactly as (Xia et al., 2022). We report the performance of the model which achieves the best validation performance during training.

### E.2 INTEGRATION WITH LARGE LANGUAGE MODELS

To incorporate the graph structural information into the language model, we use a lightweight MLP to project the 32-dimensional PSE generated by GFSE into the 4096-dimensional embedding space of LLaMA2-7B (Touvron et al., 2023). The MLP ensures alignment between the PSE and the language model's embeddings, facilitating smooth integration of graph information. We then concatenate the textual description of the central node with those of its one-hop neighbors, prepend the projected PSE as a soft token, and append a special graph token at the end of the sequence. This tokenized sequence is fed into the language model, and the hidden embedding corresponding to the special graph token is extracted to represent the central node. We use a contrastive loss between positive node pair $(i, j) \in E$ and negative node pair $(i, j') \notin E$ to train the MLP and finetune the language model

$$\mathcal{L} = \sum_{(i,j) \in E} \left( d_{ij}^2 + \max \left( \tau - d_{ij'}, 0 \right)^2 \right), \text{ with } d_{ij} = 1 - \cos(v_i, v_j), \tag{16}$$

where $v_i$ indicates the representation of the node $i$, and $\tau$ is the similarity margin (set as $0.5$ in our experiments).

## F MORE EXPERIMENTAL RESULTS

### F.1 RECONSTRUCTION OF OTHER PSE TYPES

Table 11: Performance of other PSE reconstruction on $5\%$ MolPCBA dataset. The coefficient of determination $R^2$ scores are reported as the metric.

| PSE type | ElstaticPE | LapPE | RWSE | HKdiagSE | CycleSE |
|----------|-----------|-------|------|----------|---------|
| GPSE | **0.964** | **0.973** | 0.984 | 0.981 | 0.977 |
| Ours | 0.947 | 0.970 | **0.987** | **0.984** | **0.992** |

Table 11 demonstrates that the PSE generated by our pre-trained GFSE, followed by a trainable lightweight MLP, is capable of reconstructing various pre-defined PSE types on $5\%$ MolPCBA dataset. We evaluate this using the coefficient of determination $R^2$ scores as a metric. Notably, our model performs competitively compared with GPSE, achieving an $R^2$ of 0.987 for RWSE and 0.992 for CycleSE, given the fact that GPSE directly adopts PSE reconstruction as its training objective. Instead, our method generalizes well across different PSEs without being directly trained for reconstruction. This suggests that our structural self-supervision tasks are effective and sufficient in capturing important structural information.

### F.2 DOWNSTREAM EVALUATION PERFORMANCE

We report the standard deviations of downstream performances in Table 12 and Table 13.

Table 14 compares our approach with GraphPEFT. GraphPEFT involves two steps: pre-training a graph encoder on domain-specific graph data and fine-tuning it on evaluation datasets.

Table 12: Performance on MolPCBA, ZINC (subset), Peptides-func and Peptides-struct.

| | MolPCBA AP ↑ | ZINC MAE ↓ | Peptides-func AP ↑ | Peptides-struct MAE ↓ |
|---|---|---|---|---|
| GCN | $0.2424_{\pm 0.0034}$ | $0.3670_{\pm 0.0110}$ | $0.5930_{\pm 0.0023}$ | $0.3496_{\pm 0.0013}$ |
| GCN+LapPE | $0.2417_{\pm 0.0047}$ | $0.2052_{\pm 0.0132}$ | $0.6021_{\pm 0.0051}$ | $0.2688_{\pm 0.0027}$ |
| GCN+RWSE | $0.2438_{\pm 0.0028}$ | $0.1741_{\pm 0.0528}$ | $0.5827_{\pm 0.0046}$ | $0.3270_{\pm 0.0019}$ |
| GCN+GPSE | $0.1958_{\pm 0.0074}$ | $\mathbf{0.1218}_{\pm 0.0613}$ | $0.5959_{\pm 0.0034}$ | $0.2710_{\pm 0.0041}$ |
| **GCN+GFSE** | $\mathbf{0.2477}_{\pm 0.0021}$ | $0.1237_{\pm 0.0428}$ | $\mathbf{0.6131}_{\pm 0.0074}$ | $\mathbf{0.2513}_{\pm 0.0054}$ |
| GIN | $0.2703_{\pm 0.0023}$ | $0.5260_{\pm 0.0510}$ | $0.5498_{\pm 0.0079}$ | $0.3547_{\pm 0.0045}$ |
| GIN+LapPE | $0.2701_{\pm 0.0013}$ | $0.2203_{\pm 0.0386}$ | $0.5323_{\pm 0.0083}$ | $0.2650_{\pm 0.0041}$ |
| GIN+RWSE | $0.2781_{\pm 0.0031}$ | $0.1731_{\pm 0.0614}$ | $0.5410_{\pm 0.0068}$ | $0.3282_{\pm 0.0037}$ |
| GIN+GPSE | $0.2765_{\pm 0.0073}$ | $0.2162_{\pm 0.0429}$ | $0.5389_{\pm 0.0094}$ | $\mathbf{0.2581}_{\pm 0.0046}$ |
| **GIN+GFSE** | $\mathbf{0.2839}_{\pm 0.0046}$ | $\mathbf{0.1689}_{\pm 0.0524}$ | $\mathbf{0.5532}_{\pm 0.0103}$ | $0.2674_{\pm 0.0039}$ |
| Transformer (TF) | $0.0808_{\pm 0.0117}$ | $0.6943_{\pm 0.0328}$ | $0.4800_{\pm 0.0076}$ | $0.4192_{\pm 0.0028}$ |
| TF+LapPE | $0.1784_{\pm 0.0329}$ | $0.5101_{\pm 0.0724}$ | $0.6307_{\pm 0.0091}$ | $0.2514_{\pm 0.0031}$ |
| TF+RWSE | $0.2083_{\pm 0.0674}$ | $0.2193_{\pm 0.0640}$ | $0.6326_{\pm 0.0028}$ | $0.3344_{\pm 0.0028}$ |
| TF+GPSE | $0.2040_{\pm 0.0531}$ | $0.1883_{\pm 0.0263}$ | $0.6534_{\pm 0.0041}$ | $0.2479_{\pm 0.0068}$ |
| **TF+GFSE** | $\mathbf{0.2376}_{\pm 0.0342}$ | $\mathbf{0.1548}_{\pm 0.0796}$ | $\mathbf{0.6642}_{\pm 0.0025}$ | $\mathbf{0.2436}_{\pm 0.0071}$ |
| GPS | $0.2869_{\pm 0.0045}$ | $0.1182_{\pm 0.0049}$ | $0.6535_{\pm 0.0041}$ | $0.2500_{\pm 0.0012}$ |
| GPS+LapPE | $\mathbf{0.2939}_{\pm 0.0016}$ | $0.1078_{\pm 0.0084}$ | $0.6494_{\pm 0.0037}$ | $0.2501_{\pm 0.0026}$ |
| GPS+RWSE | $0.2907_{\pm 0.0028}$ | $0.0700_{\pm 0.0040}$ | $0.6603_{\pm 0.0085}$ | $0.2739_{\pm 0.0063}$ |
| GPS+GPSE | $0.2911_{\pm 0.0036}$ | $0.0648_{\pm 0.0030}$ | $0.6688_{\pm 0.0151}$ | $\mathbf{0.2464}_{\pm 0.0025}$ |
| **GPS+GFSE** | $0.2916_{\pm 0.0061}$ | $\mathbf{0.0613}_{\pm 0.0026}$ | $\mathbf{0.6874}_{\pm 0.0120}$ | $0.2474_{\pm 0.0051}$ |
| GFSE Imp.(%) | 32.60 | 76.43 | 2.78 | 42.47 |

Table 13: Test Accuracy (%) on ogbn-arxiv, PubMed, MNIST and CIFAR10.

| | ogbn-arxiv | PubMed | MNIST | CIFAR10 |
|---|---|---|---|---|
| GateGCN | $71.69_{\pm 0.21}$ | $76.86_{\pm 0.41}$ | $97.34_{\pm 0.14}$ | $67.31_{\pm 0.31}$ |
| GateGCN+LapPE | $71.95_{\pm 0.37}$ | $74.83_{\pm 0.24}$ | $97.10_{\pm 0.28}$ | $65.08_{\pm 0.26}$ |
| GateGCN+RWSE | $71.83_{\pm 0.65}$ | $76.11_{\pm 0.39}$ | $96.84_{\pm 0.27}$ | $65.26_{\pm 0.68}$ |
| GateGCN+GPSE | $72.17_{\pm 0.42}$ | $71.97_{\pm 0.36}$ | $96.94_{\pm 0.17}$ | $65.63_{\pm 0.27}$ |
| **GateGCN+GFSE** | $\mathbf{72.61}_{\pm 0.53}$ | $\mathbf{78.39}_{\pm 0.84}$ | $\mathbf{97.44}_{\pm 0.31}$ | $\mathbf{68.39}_{\pm 0.47}$ |
| Transformer (TF) | $5.86_{\pm 0.00}$ | $\mathbf{66.63}_{\pm 0.73}$ | $97.29_{\pm 0.11}$ | $69.04_{\pm 0.28}$ |
| TF+LapPE | $5.86_{\pm 0.00}$ | $66.27_{\pm 0.46}$ | $96.95_{\pm 0.38}$ | $69.01_{\pm 0.61}$ |
| TF+RWSE | $5.86_{\pm 0.00}$ | $64.43_{\pm 0.37}$ | $97.81_{\pm 0.58}$ | $70.70_{\pm 0.45}$ |
| TF+GPSE | $21.56_{\pm 2.74}$ | $65.89_{\pm 0.14}$ | $97.78_{\pm 0.32}$ | $69.57_{\pm 0.16}$ |
| **TF+GFSE** | $\mathbf{23.84}_{\pm 3.15}$ | $66.30_{\pm 0.68}$ | $\mathbf{98.03}_{\pm 0.84}$ | $\mathbf{71.33}_{\pm 0.23}$ |
| GPS | $70.68_{\pm 0.71}$ | $\mathbf{74.26}_{\pm 0.60}$ | $98.05_{\pm 0.12}$ | $71.49_{\pm 0.35}$ |
| GPS+LapPE | $69.51_{\pm 0.38}$ | $73.68_{\pm 0.37}$ | $98.16_{\pm 0.28}$ | $71.87_{\pm 0.21}$ |
| GPS+RWSE | $72.14_{\pm 0.84}$ | $72.87_{\pm 0.44}$ | $\mathbf{98.19}_{\pm 0.30}$ | $71.30_{\pm 0.33}$ |
| GPS+GPSE | $71.21_{\pm 0.34}$ | $73.71_{\pm 0.70}$ | $98.08_{\pm 0.13}$ | $72.31_{\pm 0.25}$ |
| **GPS+GFSE** | $\mathbf{72.30}_{\pm 0.13}$ | $74.20_{\pm 0.35}$ | $98.15_{\pm 0.46}$ | $\mathbf{74.11}_{\pm 0.93}$ |
| GFSE Imp.(%) | 6.84 | 0.38 | 0.31 | 1.99 |

This process requires a domain-specific encoder for each dataset, increasing the adaptation cost when moving across different domains. In contrast, our model is designed for general-domain usage, offering a more flexible and cost-effective adaptation without requiring domain-

Table 14: Comparison with GraphPEFT

| | Cloth | | Home | | Sport | |
|---|---|---|---|---|---|---|
| | Hit@1 | MRR | Hit@1 | MRR | Hit@1 | MRR |
| GraphPEFT | 76.95 | 84.71 | 79.87 | 86.76 | 64.61 | 77.34 |
| w.o. pre-training | 76.74 | 84.57 | 79.68 | 86.63 | 64.44 | 77.21 |
| **LLaMA + GFSE** | 76.84 | 84.68 | 79.85 | 86.77 | 64.79 | 76.24 |

specific pre-training. As shown in the table, our model consistently outperforms the version of GraphPEFT that skips the domain-specific pre-training step (w.o. pre-training). This further highlights the robustness and generalizability of our approach, as it avoids the need for costly pre-training on specific domains while still achieving competitive or superior results. Specifically, for the "Sport" dataset, our model demonstrates comparable performance to GraphPEFT with pre-training, further underscoring the adaptability of GFSE in varied contexts.

## F.3 Efficiency Evaluation

During the pre-training stage on eight datasets, the average time is around 30 to 40 minutes for each epoch with a single NVIDIA A40 48GB GPU. Total training time is less than two days, which is relatively efficient for a comprehensive multi-dataset pre-training process.

We compare the inference efficiency of GFSE with handcrafted positional encodings, such as LapPE and RWSE in Table 15 and Table 16. Specifically, we generate 1,000 synthetic Erdos-Rényi graphs for various graph sizes (100, 300, 500, and 1,000 nodes) and evaluate the time required for pre-computation and inference in GFSE.

As shown in the table, both LapPE and RWSE exhibit significant increases in computation time as the graph size grows. Pre-computation times required by GFSE inference remain minimal for all graph sizes, underlining

Table 15: Runntimes ($s$) of PSE computation on random synthetic graph with increasing numbers of nodes

| PSE / Graph size | 100 | 300 | 500 | 1000 |
|---|---|---|---|---|
| LapPE | 2 | 9.25 | 34 | 155 |
| RWSE | 2 | 9.76 | 31.48 | 207 |
| Pre-computation | 0.0007 | 0.001 | 0.003 | 0.006 |
| GFSE Inference | 0.908 | 3.958 | 10.770 | 48.106 |

Table 16: Runntimes ($s$) on real-world graph dataset

| Dataset | ZINC-subset | MolHIV | MolPCBA | Peptides | MNIST | CIFAR10 |
|---|---|---|---|---|---|---|
| LapPE | 25 sec | 37 sec | 6.13 min | 73 sec | 96 sec | 2.55 min |
| RWSE | 11 sec | 58 sec | 8.33 min | - | - | - |
| GFSE | 4.17 sec | 17.23 sec | 2.97 min | 15.21 sec | 49.38 sec | 1.27 min |

the model's efficiency in this phase. In Table 16, we observe that GFSE demonstrates superior scalability in inference, making it a more efficient option for large-scale graph processing.

## G Discussion

**Limitation and Social Impact**. While GFSE represents a significant step forward in developing general and expressive foundation models for graph-structured data, there are certain limitations to consider. For example, the effectiveness of GFSE may be influenced by the diversity and quality of the pre-training datasets, as biases or under-representation in the data could propagate to the learned representations. Future work could focus on expanding the diversity and scale of pre-training datasets to mitigate such biases and improve the robustness of GFSE across more diverse domains

From a social impact perspective, the development of structural graph foundation models like GFSE holds significant promise in advancing various application domains that rely on graph analytics. In fields such as computational biology, social network analysis, and recommendation systems, GFSE could enable more accurate and efficient modeling of complex structured data, leading to improved understanding and decision-making. Furthermore, by reducing the need for extensive task-specific fine-tuning, GFSE could democratize the use of powerful graph learning techniques, making them more accessible to researchers and practitioners with limited computational resources.

