# OpenReview forum: "GFSE: A Foundational Model For Graph Structural Encoding"
_ICLR.cc/2025/Conference — Submitted to ICLR 2025_

### Official Review · Reviewer_5nqo · 2024-10-27

**Soundness:** 2
**Presentation:** 3
**Contribution:** 2
**Rating:** 5
**Confidence:** 4

**Summary:**

The authors propose a universal graph position encoding method, GFSE, designed for graphs from different domains. The method utilizes a graph transformer as its backbone and is pretrained with four self-supervised learning tasks—such as shortest path prediction, motif counting, and contrastive learning—to capture structural knowledge. Experimental results demonstrate the method’s effectiveness across diverse tasks and domains.

**Strengths:**

1. The paper is well-written and easy to follow.

2. The proposed method is supported by a solid theoretical foundation, demonstrating its effectiveness.

3. The method is versatile and can be applied to various tasks, including basic graph reasoning, classification-related tasks, and LLM-based inference.

**Weaknesses:**

1. The contribution of the proposed approach is unclear, as it does not appear to be a graph foundation model applicable for inference across various graphs. Instead, it may be more accurately described as a universal graph positional encoding method.

2. The motivation for adopting a universal positional embedding is unclear. Couldn’t we simply train a positional encoder for each dataset individually? Additionally, the paper lacks evidence that the model pretrained on one dataset (e.g., Dataset A) can be successfully transferred to another dataset (e.g., Dataset B) with strong performance.

3. While the authors present experimental results on graph and node classification, it would be interesting to see if this method significantly improves performance on link prediction tasks and on heterophilic graphs, where capturing structural insights is especially important.

**Questions:**

1. The motivation behind GFSE appears somewhat similar to UniAug [1], both aiming to identify universal structural patterns. Could the authors elaborate on the differences between these approaches?

2. The authors have not provided an official codebase, which may limit the practical usability of the method. Do the authors plan to release the code?

3. Can the proposed method be applied effectively in scenarios with limited label availability?

Reference:

[1] Cross-Domain Graph Data Scaling: A Showcase with Diffusion Models, 2024.

---

> ### Author Response · Authors · 2024-11-24
> **Response to reviewer 5nqo (1/2)**
>
> Thanks for your valuable feedback and positive comments.  We address the potential concerns as follows.
>
> > Q1: The contribution of the proposed approach is unclear. It does not appear to be a graph foundation model, but a universal graph positional encoding method.
>
> We agree that it may be more appropriate to frame this work as a universal structural encoding model. However, achieving a full-fledged GFM, capable of seamless adaptation across all domains, remains a significant challenge. In this work, we focus on a **different but complementary** goal—the exploration of universal structural patterns in graphs, which we believe are critical in developing a GFM. Our GFSE is not intended as an independent GFM, but rather as an assistant to enhance the structural awareness of graph models, whether train-from-scratch models or pre-trained GFMs. Rather than positioning our work as a full-fledged GFM for diverse graph tasks, the main contribution of the paper is to verify the effectiveness and potential of structural patterns in improving the generalizable representation of graph distributions.
>
> [1]Towards Graph Foundation Models: A Survey and Beyond
>
> > Q2: The motivation for adopting a universal positional embedding is unclear.
>
> We discuss the motivation of universal positional embedding in Line 49~53. The central idea is tied to the broader goal of developing a graph foundation model (GFM). To achieve this, it is essential to identify universal attributes—domain-agnostic features that can be applied across different graph domains. In this context, we draw inspiration from NLP, where the concept of tokens and contexts is foundational. Similarly, in graph learning, we observe that structural patterns (such as motifs, node connectivity, and graph topology) can also be shared and transferred across different graph domains. This insight motivates the design of a universal positional encoder—a component that captures these transferable structural patterns and makes them available to a wide variety of graph models.
>
> > Q3: The paper lacks evidence that the model pre-trained on one dataset (e.g., Dataset A) can be successfully transferred to another dataset (e.g., Dataset B) with strong performance.
>
> We have indeed tested the generalizability of our GFSE in **Table 3** and **Table 4**, where datasets like Arxiv, MNIST, and molecular datasets from the OGB benchmark were not part of the pre-training set. Despite this, our experiments show that GFSE consistently outperforms traditional PSE such as RWSE and LapPE. Furthermore, the GFSE outperforms GPSE, which was pre-trained specifically on a single dataset, highlighting the importance of cross-domain pre-training for improved generalizability to unseen datasets. We believe these results strongly support the claim that our GFSE is capable of transferring knowledge learned from one domain to another, and we will revise the manuscript to make this point clearer.
>
> >  Q4: It would be interesting to see if this method significantly improves performance on link prediction tasks and on heterophilic graphs.
>
> We kindly bring to your consideration that **the experiments presented in Section 4.6 indeed focus on link prediction**. Specifically, the downstream task we evaluate is the prediction of whether two items will be co-purchased in a recommendation system. The performance demonstrates the effectiveness of pre-trained PSE in providing structural patterns that are crucial for edge-level tasks.
>
> **Our current experiments are not limited to homophilic graphs, where molecular graphs also show a certain heterophilic level**. Given the limited time during the rebuttal process, we were unable to include additional experiments on extremely heterophilic graphs. However, we plan to extend our evaluation to include these scenarios in future work and provide a more comprehensive analysis of the model’s capabilities.

---

> > ### Author Response · Authors · 2024-11-24
> > **Response to reviewer 5nqo (2/2)**
> >
> > > Q5: Could the authors elaborate on the differences from UniAug?
> >
> > While both approaches aim to identify universal structural patterns, there are several fundamental differences in both methodology and objectives:
> >
> > - **Core Objective Distinction**: GFSE focuses on learning generalizable structural representations that can be directly used across different downstream tasks, while UniAug primarily serves as a data augmentation tool, aiming to generate or refine graph structures specifically tailored to downstream tasks
> > - **Methodological Differences**: GFSE emphasizes learning an expressive and powerful encoder that captures intrinsic graph properties without task-specific guidance, while UniAug relies on diffusion and guidance head, which may lead to task-specific biases and reduced generalizability
> > - **Downstream Application**: UniAug requires generating new synthetic structures for each task using specific guidance objectives, while GFSE provides ready-to-use structural encodings that can be immediately applied to any downstream task
> > We believe these distinctions highlight that they serve fundamentally different purposes and operate under different paradigms.
> >
> > >  Q6: Do the authors plan to release the code?
> >
> > Yes, we will have an official codebase once acceptance. We also included a temporary codebase at https://anonymous.4open.science/r/GFSE-E8C0 for review reference.
> >
> >
> > > Q7: Can the proposed method be applied effectively in scenarios with limited label availability?
> >
> > We would like to emphasize that our proposed method is well-suited for scenarios with limited label availability. One of the key advantages is that the pre-training tasks are self-supervised, meaning they do not require any annotations or labels. We employ self-supervised learning techniques such as structure reconstruction and contrastive learning to train the model. These tasks allow the model to capture useful structural patterns in the graph without the need for manually labeled data. Once pre-trained, the GFSE can also be fine-tuned on downstream tasks with limited labels, where it leverages the structural awareness learned during pre-training to improve performance.

---

> > > ### Comment · Reviewer_5nqo · 2024-11-25
> > >
> > > Thank you for the detailed clarifications. While some concerns have been addressed, several major points remain unresolved:
> > >
> > > > 1: Regarding Q1 and Q2:
> > > >
> > > I agree that understanding structural information is critical in designing graph foundation models. However, I disagree with achieving this by designing a plug-and-play module to supplement a graph foundation model. Instead, the graph foundation model itself should possess the ability to identify motifs or substructures within graphs. For comparison, in the design of language foundation models (e.g., LLMs) or visual foundation models (e.g., VLMs), the models intrinsically understand contextual information without requiring additional modules to achieve this. Consequently, I cannot fully agree with the statement: "We focus on a different but complementary goal—the exploration of universal structural patterns in graphs." Instead, these capabilities should be integral to the model itself.
> > >
> > > > 2. Regarding Q3:
> > > >
> > > While the authors claim that datasets like Arxiv, MNIST, and molecular datasets from the OGB benchmark were not part of the pre-training set, the manuscript (lines 314–315) indicates otherwise: "We utilize a diverse collection of cross-domain datasets for pretraining, ensuring a broad spectrum of graph structures and scales, including MolPCBA, MolHIV, MNIST, peptides, ogbn-proteins, Pokec, ogbn-arxiv, and ogbn-product." These datasets are explicitly mentioned as part of the pre-training set. Additionally, it is natural for graphs within the same domain to exhibit similar patterns. I noticed that the testing graphs belong to the same domains as those in the pre-training set. Can the authors provide results demonstrating that a model pre-trained on domain A achieves strong performance on domain B? This would strengthen the contribution of the work, as it is impractical to include all types of graphs during pre-training.
> > >
> > > > 3. Regarding Q4:
> > > >
> > > Section 4.6 provides results using LLMs for inference. However, the authors should include experimental results without relying on LLMs, as their strong general knowledge could overshadow the structural embedding's contribution to link prediction. This makes it difficult to isolate the benefits of the proposed structural embedding for the task.
> > >
> > > Regarding experiments on heterophilic graphs, the statement, "Our current experiments are not limited to homophilic graphs, where molecular graphs also show a certain heterophilic level," is insufficient. Heterophily impacts graph classification and node classification differently, and it is crucial for a structural embedding method to evaluate performance on highly heterophilic graphs. While I understand the time constraints, I strongly recommend conducting experiments on extremely small heterophilic graph datasets, such as the well-known Texas, Wisconsin, and Cornell graphs. These would provide valuable insights without requiring extensive computational resources.

---

> > > > ### Author Response · Authors · 2024-12-03
> > > >
> > > > Thanks for your continued feedback.
> > > >
> > > > > Regarding Q1 and Q2
> > > >
> > > > We would like to clarify that **GFSE is designed as a foundation structural encoder, not as a standalone graph foundation model**. Our work highlights that, in practice, learning universal structural patterns as part of pre-training **is not easily achieved in isolation**. However, as we show, GFSE provides a **scalable and flexible approach** to enable graph models to learn these patterns across diverse domains, without requiring specialized domain-specific designs. We would like to point out that the expectation to **generalize universally across all possible feature spaces—essentially seeking AGI-level performance—is unrealistic for current models, including ours**. Our model is not designed to handle every arbitrary feature space but to advance the transferability of graph representations across a wide variety of domains, which is a much more practical and relevant goal in graph learning.
> > > >
> > > > > Regarding Q3
> > > >
> > > > Thanks for bringing this to our attention. We would like to clarify molecular datasets from the OGB benchmark are not in the pre-training set. To further demonstrate cross-domain generalizabiltiy, we did the following experiments during the rebuttal process, where we compared with the model pre-trained on MolPCBA only and trained from scratch in downstream application, separately.
> > > >
> > > > |                                 | MolPCBA (↑) | ZINC(↓)    | Arxiv(↑)  |
> > > > |---------------------------------|-------------|------------|-----------|
> > > > | 1. pre-trained on all domains | **0.2916**  | **0.0613** | **72.30** |
> > > > | 2. pre-trained on MolPCBA          | 0.2914      | 0.0638     | 71.24     |
> > > > | 3. train-from-scratch              | 0.2910      | 0.0724     | 71.85     |
> > > >
> > > > We observe that
> > > > - **Pre-training is Crucial for Better Graph Representations (1, 2, 3)**: When comparing the performance of GFSE trained-from-scratch V.S. GFSE pre-trained on diverse domains, we observe that pre-training leads to better downstream performance in most cases. This underscores the importance of pre-training for generating more expressive and beneficial graph representations.
> > > >
> > > > - **GFSE essentially learns cross-domain structural patterns (1,3)**: Additionally, we observe that when GFSE is trained from scratch on a singular dataset, the downstream performance is consistently lower than when the model is pre-trained on multiple domains. This suggests that cross-domain pre-training is a crucial factor for improving performance.
> > > >
> > > > > Regarding Q4
> > > >
> > > > We will conduct additional experiments that focus purely on the structural embedding's contribution, without the use of LLMs, to isolate the impact of the graph-specific representation, and on heterophilic graphs as well.

---

### Official Review · Reviewer_jHor · 2024-11-02

**Soundness:** 2
**Presentation:** 3
**Contribution:** 2
**Rating:** 3
**Confidence:** 5

**Summary:**

This paper proposes GFSE (Graph Foundational Structural Encoder), a graph transformer model pre-trained on diverse graph datasets using multiple self-supervised tasks to generate positional and structural encodings (PSE). This model aims to serve as a universal structural encoder that can enhance various downstream graph learning tasks and models. The key contributions include: (1) a multi-task pre-training framework with four structural tasks, (2) some theoretical analysis of the model's expressiveness, and (3) some empirical validations across different graph domains and model architectures.

**Strengths:**

1. This paper is generally easy to follow and well-structured.
2. The multi-task pre-training approach combining different structural aspects is interesting.
3. The empirical performance shows some gain.

**Weaknesses:**

1. **Overclaiming and Overstated Results**:
   - The claim of being "the first cross-domain graph structural encoder" ignores relevant prior work:
     * GCC [1] already proposed cross-domain pre-training
     * GraphMAE [2] and other self-supervised approaches [3,4] have demonstrated cross-domain capabilities
   - The "20.48% average improvement" appears selectively calculated:
     * Tables 2-3 show many improvements <1% (e.g., MNIST: 0.31%, PubMed: 0.38%)
     * Best results seem cherry-picked from different model combinations

2. **Limited Technical Novelty**:
   - The core architecture is largely borrowed from GPS [5] with minimal modifications:
     * The biased attention mechanism is a straightforward extension of existing work [6]
     * The pre-training tasks are mostly adapted from prior graph learning literature [7,8]
   - The multi-task learning setup uses standard techniques:
     * Uncertainty-based loss weighting is directly from [9]
     * Community detection approach follows [10]
     * Motif counting implementation is based on [11]

3. **Methodological Limitations**:
   - Pre-training effectiveness:
     * No comparison with recent advances in cross-domain graph pre-training [12,13]
     * Missing analysis of task interactions shown to be crucial in [14]
   - Theoretical guarantees:
     * The expressiveness analysis follows similar arguments to [15]
     * The proofs rely on assumptions challenged in recent work [16]

[1] Qiu et al., "GCC: Graph Contrastive Coding for Graph Neural Network Pre-training", KDD 2020

[2] Hou et al., "GraphMAE: Self-supervised Masked Graph Autoencoders", KDD 2022

[3] You et al., "Graph Contrastive Learning with Augmentations", NeurIPS 2020

[4] Xu et al., "Self-supervised Graph-level Representation Learning with Local and Global Structure", ICML 2021

[5] Rampášek et al., "Recipe for a General, Powerful, Scalable Graph Transformer", NeurIPS 2022

[6] Dwivedi et al., "Graph Neural Networks with Learnable Structural and Positional Representations", ICLR 2022

[7] Veličković et al., "Deep Graph Infomax", ICLR 2019

[8] Hu et al., "Strategies for Pre-training Graph Neural Networks", ICLR 2020

[9] Kendall et al., "Multi-task Learning Using Uncertainty to Weigh Losses", CVPR 2018

[10] Blondel et al., "Fast Unfolding of Communities in Large Networks", Journal of Statistical Mechanics 2008

[11] Bouritsas et al., "Improving Graph Neural Network Expressivity via Subgraph Isomorphism Counting", TPAMI 2022

[12] Liu et al., "Graph Neural Networks: Foundations, Frontiers, and Applications", 2023

[13] Davies et al., "Towards Generalised Pre-training of Graph Models", 2024

[14] Sun et al., "Multi-task Self-supervised Learning for Graph Neural Networks", AAAI 2023

[15] Zhu et al., "On Structural Expressive Power of Graph Transformers", KDD 2023

[16] Maron et al., "On the Universality of Graph Neural Networks on Large Random Graphs", NeurIPS 2023

**Questions:**

See weakness

---

> ### Author Response · Authors · 2024-11-24
> **Response to reviewer jHor**
>
> Thanks for your valuable feedback and positive comments.  We address the potential concerns as follows.
>
> > Q1: The claim of being "the first cross-domain graph structural encoder" ignores relevant prior work.
>
> We would like to clarify that GFSE is the first **cross-domain** graph structural encoder that is supervised by **multiple pre-training tasks**. We kindly emphasize that the term "cross-domain" in our work refers to the model's ability to generate useful representations for graphs across different domains, without being limited to any specific type of graph data.
>
> While GCC does propose cross-domain pre-training, it relies on a singular task, specifically graph contrastive learning, which leads to relatively coarse-grained embedding and suboptimal performance. In contrast, GraphMAE and other self-supervised methods focus on domain-specific encoders, such as those tailored for molecular graphs, which constrains their ability to generalize to different domains.
>
> Our GFSE goes beyond these limitations by incorporating multiple pre-training tasks—including structure reconstruction and contrastive learning—which enables it to learn more comprehensive and transferable structural patterns across domains.
>
> > Q2: The "20.48% average improvement" appears selectively calculated.
>
> The 20.48% improvement is **strictly averaged across eight datasets** from different domains. GFSE consistently outperforms the baseline models, especially compared to GPSE. The following table shows the improvement rate on different datasets achieved by GPSE and our GFSE, which shows the significance of our contribution.
>
> |      | MolPCBA | ZINC  | Peptides-func | Peptides-struct | Arxiv | PubMed | MNIST | CIFAR10 |
> |------|---------|-------|---------------|-----------------|-------|--------|-------|---------|
> | GPSE | 16.16   | 73.02 | 0.37          | 41.48           | 5.57  | -2.06  | 0.04  | -0.11   |
> | GFSE (ours) | 32.60   | 76.43 | 2.78          | 42.47           | 6.84  | 0.38   | 0.31  | 1.99    |
>
> It is important to note that the smaller improvements observed in some datasets (such as MNIST and PubMed) are due to these graphs **relying less on structural information**.
>
>
> > Q3: Limited Technical Novelty: The core architecture is largely borrowed from GPS with minimal modifications and the multi-task learning setup uses standard techniques.
>
> While it is true that we draw inspiration from GPS and related works, we believe that the core innovation here is not simply in how these modifications are integrated to achieve theoretically proven powerful expressivity, but in how we leverage multi-task learning to capture universal structural patterns across diverse graph domains, which is a key distinguishing feature from GPS. Moreover, we would like to emphasize that the main contribution of our manuscript is an attempt to investigate the effectiveness of structural patterns as universal attributes for graph foundational modeling. It serves as an assistant to enhance the structural awareness of graph models, whether train-from-scratch models or pre-trained GFMs.
>
>
> > Q4: No comparison with recent advances in cross-domain graph pre-training
>
> We indeed discuss the method in [13] in the Related Work (Line 116). The approach in [13] uses a singular pre-training task and a relatively simple GNN encoder, which results in suboptimal embedding quality. The method in [13] is not open-source yet. Therefore, we didn’t involve this approach as a baseline in this submission. We acknowledge that future work could benefit from a direct empirical evaluation of these models.
>
> > Q5: Missing analysis of task interactions shown to be crucial.
>
> We have provided insights on task interactions in **Section 4.7**, where we analyze the impact of each pre-training task on the overall model performance. In this section, we demonstrate how different tasks contribute to the model’s ability to capture both local and global structural patterns. We also discuss the trade-offs between tasks and how they affect the quality of the learned representations.
>
> > Q6: Theoretical guarantees
>
> While our expressiveness analysis shares some foundational elements with [15], we meaningfully extend these results to the random walk positional encoding. We acknowledge the challenges raised in [16] regarding our assumptions. However, it is important to highlight that [16] primarily focuses on local GNN-based architectures, which are fundamentally different from the global structural focus of our framework. Moreover, our theoretical findings are strongly supported by empirical results, which are not included in [16].  We would like to emphasize that while we borrow some insights from previous studies, we have adapted them to our global transformer architecture based on random walk positional encoding, highlighting the theoretical soundness and empirical validity of the proposed framework.

---

> > ### Comment · Reviewer_jHor · 2024-11-25
> >
> > 1. I don't agree that "the first cross-domain graph structural encoder that is supervised by multiple pre-training tasks" is an actual contribution and the motivation behind this claim is really weak.
> > 2. Your average performance improvement is also an overstatement, I don't think such calculation is fair since your improvements have a high variance.
> > 3. If your motivation comes only from empirical results, then your contribution is further weakened since "universal structural patterns" learned from such multi-task learning are not clearly investigated.
> >
> > Based on these assessments, I'm not satisfied with the rebuttal and therefore I will keep the scores.

---

> > > ### Author Response · Authors · 2024-12-03
> > >
> > > Thank you for your feedback.
> > > > Regarding the contribution of “first cross-domain graph structural encoder supervised by multiple pre-training tasks”
> > >
> > > We respectfully disagree with the assessment that our contribution is weak or unsubstantiated. We clarify that this claim is based on a **fundamental shift in paradigm**, moving beyond the limitations of **domain-specific single-aspect graph pre-training** that has dominated the graph representation learning field. No existing model, to our knowledge, has achieved this level of generalizability across graph domains, where the learned representation can successfully transfer to different tasks without being overfitted to a single domain.
> > >
> > > > Regarding the performance improvement
> > >
> > > We acknowledge the presence of variance in downstream tasks, which is common when working with diverse datasets. However, the key point is that **GFSE consistently outperforms** traditional pre-trained models and even models trained on domain-specific data, demonstrating substantial improvements in terms of generalization. We calculated the performance improvement as an average across multiple datasets, which is a fair reflection of its effectiveness across different graph domains. **The variance you mention should not overshadow the overall trend of better performance** when leveraging the cross-domain pre-training strategy.
> > >
> > > > Regarding the motivation behind our approach
> > >
> > > We strongly assert that the claim of learning universal structural patterns is not merely driven by empirical results but is deeply rooted in the theoretical framework of graph representation learning. Structural patterns like small-world properties, community structures, and scale-free distributions are well-documented and commonly observed across multiple graph domains. The challenge we address is not just to collect these patterns but to design a unified model that can learn them effectively without being bound to a single graph domain. This is a novel contribution that has broad implications for graph model generalization.
> > > Finally, we would like to point out that **the expectation to generalize universally across all possible feature spaces—essentially seeking AGI-level performance—is unrealistic** for current graph foundation models, including ours. **Our model is not designed to handle every arbitrary feature space** but to advance the transferability of graph representations across a wide variety of domains, which is a much more practical and relevant goal in graph learning. We firmly believe that this is a meaningful step forward in the development of more generalizable graph models.

---

### Official Review · Reviewer_g26S · 2024-11-03

**Soundness:** 2
**Presentation:** 2
**Contribution:** 2
**Rating:** 5
**Confidence:** 3

**Summary:**

To facilitate pre-trained graph models across diverse domains, this paper proposes a cross-domain graph structural encoder, GFSE. This encoder is built on a graph transformer architecture and is pre-trained on multiple domains with multiple self-supervised learning objectives. After pre-training, the encoder is evaluated on two types of downstream graphs, i.e., vectorized-feature graphs and text-attributed graphs.


Thank you for providing a detailed rebuttal. After reviewing your responses and the revised manuscript, my concerns regarding the technical novelty remain unresolved. Additionally, the revised paper still does not include comparisons with state-of-the-art baseline models.
As such, I have decided to maintain my original score.

**Strengths:**

1. The paper is well-organized and easy to follow.
2. The proposed method is technically sound, which introduces multiple self-supervised learning objectives to pre-train a generalizable structural encoder.
3. The paper provides a complexity analysis and compares the runtime with some existing structural encodings.

**Weaknesses:**

1. The proposed model is not very novel, as most components have been introduced in prior graph transformer models.
2. There is a lack of detailed descriptions of the method. For example, when encoding graphs from different domains using the same encoder, how are the graphs featurized to ensure they can be encoded by the same encoder?
3. The baselines used for comparison are not up-to-date. Since GPS, newer graph transformer models have been proposed, and these should also be included in the comparison.
4. The paper only conducts ablation study on different pre-training sub-objectives. It is recommended to also perform ablation study on different pre-training domains or datasets, as this could reveal which domains contribute to learning cross-domain general patterns.

**Questions:**

Please refer to the Weaknesses.

---

> ### Author Response · Authors · 2024-11-24
> **Response to reviewer g26S**
>
> Thanks for your valuable feedback and positive comments.  We address the potential concerns as follows.
>
> > Q1: The proposed model is not very novel.
>
> While it is true that the GFSE incorporates concepts from existing techniques, the key novelty in the proposed architecture lies in how these components are integrated in a creative manner to achieve theoretically proven powerful expressivity, and how they address the challenge of cross-domain graph structure pre-training. We would like to emphasize that the main contribution of our manuscript is an attempt to investigate the effectiveness of structural patterns as universal attributes for graph foundational modeling. It serves as an assistant to enhance the structural awareness of graph models, whether train-from-scratch models or pre-trained GFMs.
>
>
> > Q2: How are the graphs from different domains encoded by the same encoder?
>
> Here is some misunderstanding. We would like to clarify that GFSE only uses the graph structure as input, represented as random walk encoding in Equation (1). Once GFSE is pre-trained on the graph structure, the raw graph features can be augmented by the PSE generated by GFSE and processed by a downstream graph model, which is consequently trained on a certain dataset for task-specific capacity.
>
> > Q3: The baselines used for comparison are not up-to-date.
>
> We follow the baseline setting in GPSE [1] for evaluation and select representative MPNN and graph transformer (i.e., GPS) as the base model. We are currently in the process of incorporating several state-of-the-art graph transformer models into our downstream evaluation, such as Nodeformer and GRIT. We will include the additional experiment results in our revised version accordingly.
>
> [1] Graph Positional and Structural Encoder
>
> > Q4: It is recommended to also perform ablation studies on different pre-training domains or datasets.
>
> |                                 | MolPCBA (↑) | ZINC(↓)    | Arxiv(↑)  |
> |---------------------------------|-------------|------------|-----------|
> | 1. pre-trained on 100% all domains | **0.2916**  | **0.0613** | **72.30** |
> | 2. pre-trained on 50% all domains  | 0.2913      | 0.0658     | 72.13     |
> | 3. pre-trained on 10% all domains  | 0.2897      | 0.0703     | 71.76     |
> | 4. pre-trained on MolPCBA          | 0.2914      | 0.0638     | 71.24     |
>
> Thanks for your suggestions. To address your concern, we have carefully analyzed the impact of the number of pre-training datasets on downstream performance, and the results are summarized in the above table (the base model is GPS). Our findings show a few key trends:
> - **Performance Tends to Decrease as Pre-training Datasets Are Reduced (1-3)**: As the number of pre-training datasets decreases, we observe a slight reduction in downstream performance, indicating that a more diverse range of pre-training data can help improve the robustness of the model. This trend shows that the model benefits from exposure to a broader variety of graph structures, enhancing its ability to generalize across different graph domains.
> - **Cross-domain Pre-training is Important (1,4)**: Additionally, we observe that when GFSE is pre-trained on a singular dataset (e.g., MolPCBA), the downstream performance is consistently lower than when the model is pre-trained on multiple domains. This suggests that cross-domain pre-training is a crucial factor for improving performance.
>
> We will include complete experimental results in our revised version to verify the above conclusion.

---

### Official Review · Reviewer_h6Am · 2024-11-03

**Soundness:** 2
**Presentation:** 3
**Contribution:** 2
**Rating:** 6
**Confidence:** 3

**Summary:**

The paper introduces GFSE, a novel model designed to enhance the performance of graph-based machine learning by addressing the limitations of existing graph pre-training models. GFSE leverages a Graph Transformer architecture with biased attention mechanisms, incorporating multiple self-supervised learning objectives to capture complex, multi-level structural patterns universally across domains. This allows it to produce generic, expressive PSE that enhance downstream tasks in various graph domains, including molecular structures, social networks, and citation networks. Importantly, the results discussed in section 4.5 of the paper is particularly promising.

**Strengths:**

1. GFSE successfully addresses the challenge of domain-specificity in graph pre-training by identifying and encoding universal structural patterns.
2. By focusing on universal graph characteristics, GFSE potentially reduces the need for extensive domain-specific fine-tuning, facilitating easier deployment and adaptation in various applications.
3. The paper presents comprehensive experimental results.

**Weaknesses:**

1. While the paper introduces GFSE as an innovative architecture, it primarily appears to be a composite of existing methods such as GraphGPS, GRIT’s RRWP, and Attention Bias. The real novelty claimed, addressing domain-specificity in graph pre-training, is not compellingly validated by the experiments.

2. The downstream evaluation in section 4.4 does not include comparisons with SOTA models. The results presented do not meet the results of current SOTA models.

3. The experiments in section 4.3 lack detailed descriptions of hyperparameter tuning, which undermines the credibility of the results.

4. There is a noticeable lack of ablation studies comparing GFSE with GRIT’s RRWP, which could provide critical insights into the unique contributions and improvements made by GFSE’s specific features.

5. Missing related work such as [1] on shortest-path distance PSE and [2,3] on graph contrastive learning.

   [1] Enhancing Graph Transformers with Hierarchical Distance Structural Encoding.

   [2] Graph Contrastive Learning Automated.

   [3] AutoGCL: Automated Graph Contrastive Learning via Learnable View Generators.

**Questions:**

Does the dataset used in the experiments of section 4.6 overlap with those appearing in InstructGLM? It is recommended that the authors include the original datasets from InstructGLM.

---

> ### Author Response · Authors · 2024-11-24
> **Response to reviewer h6Am (1/2)**
>
> Thanks for your valuable feedback and positive comments.  We address the potential concerns as follows.
>
> > Q1: GFSE primarily appears to be a composite of existing methods. Addressing domain-specificity in graph pre-training, is not compellingly validated by the experiments.
>
> While it is true that the GFSE incorporates some concepts from existing techniques, the key novelty lies in how these components are integrated to achieve theoretically proven powerful expressivity in a creative manner, and how they address the challenge of domain-specificity in graph pre-training. We kindly bring to your consideration that the main contribution of our manuscript is an attempt to investigate the effectiveness of structural patterns as universal attributes for graph foundational modeling. It serves as an assistant to enhance the structural awareness of graph models, whether train-from-scratch models or pre-trained GFMs. In experiments, the superiority of GFSE over GPSE, which is pre-trained on a specific dataset, verifies the benefit and importance of cross-domain pre-training.
>
> > Q2: The downstream evaluation in section 4.4 does not include comparisons with SOTA models.
>
> Our GFSE is primarily positioned as a foundational structural encoder, we thereby have chosen to compare against traditional and pre-trained PSE. We include GPSE as a baseline, which is the most recent SOTA method for graph PSE. Additionally, we compare with other competitive PSE. GFSE has achieved larger performance improvements on average compared with GPSE (shown below), which shows the significance of our contribution. Please note that when the best performance approaches near-perfect scores for benchmark datasets, designing a generic PSE that results in a consistently better performance becomes extremely challenging. However, despite these challenges, GFSE has resulted in a performance gain that is higher than the performance gain of earlier studies.
>
> The average performance gain of GPSE (ICML2024) and Ours on different datasets:
> | Imp.(%) | MolPCBA | ZINC  | PubMed | CIFAR10 |
> |---------|---------|-------|--------|---------|
> | GPSE    | 16.16   | 73.02 | -2.06  | -0.11   |
> | Ours    | 32.60   | 76.43 | 0.38   | 1.99    |
>
> > Q3: The experiments in section 4.3 lack detailed descriptions of hyperparameter tuning
>
> Thanks for bringing this to our attention. We follow the GPS [1] framework to implement the base models, including GIN, Transformer and GPS. Specifically, the hyper-parameters of the Transformer (for triangle) and GPS (for Pattern and Cluster) are shown below. We use 2 linear layers in MLP and 6 message-passing layers in GIN. We will revise the manuscript accordingly to include more details about hyperparameter tuning.
>
> |          | # layers | # heads | Hidden dim | Dropout | Attention dropout | Batchsize | Learning Rate |
> |----------|----------|---------|------------|---------|-------------------|-----------|---------------|
> | Triangle | 6        | 4       | 64         | 0       | 0.1               | 32        | 0.001         |
> | Pattern  | 6        | 4       | 64         | 0       | 0.5               | 32        | 0.0005        |
> | Cluster  | 16       | 8       | 48         | 0.1     | 0.5               | 16        | 0.0005        |
>
>
> [1] Recipe for a General, Powerful, Scalable Graph Transformer
>
> > Q4: Ablation studies comparing GFSE with GRIT’s RRWP.
>
> |              | Arxiv     | PubMed    | MNIST     |
> |--------------|-----------|-----------|-----------|
> | GateGCN+GFSE | **72.61** | **78.39** | 97.44     |
> | GateGCN+RRWP | 71.83     | 76.53     | 96.42     |
> | GPS+GFSE     | 72.30     | 74.20     | **98.15** |
> | GPS+RRWP     | 72.25     | 74.20     | 98.05     |
>
> To address your concern, we have conducted ablation studies comparing GFSE and RRWP on three datasets using both GateGCN and GPS, as shown in the table above, The results clearly demonstrate that GFSE consistently outperforms RRWP, particularly for base models that lack global graph structures, emphasizing the strengths of our approach:
> - Our GFSE provides a more **generalizable representation compared to RRWP**, regardless of whether the model is based on a global transformer architecture or a local GNN architecture.
> - The more significant improvement when combining GFSE with GateGCN can be attributed to the fact that **GFSE incorporates global structural information**, which complements the local message-passing used by GNNs like GateGCN.
> - The embedding generated by GFSE is inherently **derived from the insights of RRWP** but is enhanced by the ability to incorporate more global structural awareness. This makes GFSE more flexible and capable of encoding universal structural patterns that are transferable across different domains. We will provide an additional comparison with RRWP in our revised version.

---

> > ### Author Response · Authors · 2024-11-24
> > **Response to reviewer h6Am (2/2)**
> >
> > > Q5: Missing related work.
> >
> > Thank you for bringing this to our attention. We will add all the missing related work in our revised version.
> >
> > > Q6: Does the dataset used in the experiments of section 4.6 overlap with those appearing in InstructGLM?
> >
> > No, the dataset used in Section 4.6 is different from those used in InstructGLM. We have performed additional experiments on the ogbn-arxiv and cora, the original datasets used by instructGLM to verify its effectiveness. We report the performance as follows:
> >
> > |                 | ogbn-Arxiv | Cora  |
> > |-----------------|------------|-------|
> > | Finetuned Llama | 74.94      | 79.95 |
> > | InstructGLM     | 75.70      | 87.08 |
> > | Llama+RRWP     | 76.03      | 90.27 |
> > | Llama+GFSE      | 76.10      | 91.26 |
> >
> > We observe that Llama+GFSE consistently outperforms InstructGLM and Llama+RRWP. The results demonstrate that (1) explicit incorporation of graph structural encoding into LLM architecture helps to improve the model’s performance in graph tasks, and (2) GFSE generates more effective and useful PSE for LLM architecture to understand the graph structure.

---

> > > ### Comment · Reviewer_h6Am · 2024-12-02
> > >
> > > Thank you for presenting a detailed rebuttal. I appreciate the additional experiments, but after carefully reviewing the other reviewers' comments, there are still some concerns that I think are unanswered.
> > >
> > > 1. It is essential to provide more experiments about what transferable information the model learns. In other words, the paper lacks evidence showing that a model pre-trained on one dataset can successfully transfer to another dataset and achieve good performance. There might be a few experiments on Arxiv and MNIST, but in these datasets, the gap with RRWP is not significant.
> > >
> > > 2. Could you please explain how Llama+RRWP or Llama+GFSE is implemented? If possible, could you provide the code for this experiment?

---

> > > > ### Author Response · Authors · 2024-12-03
> > > >
> > > > Thanks for your follow-up feedback.
> > > >
> > > > We would like to highlight that we have already conducted experiments to evaluate the generalizability of the pre-trained model across different datasets. We did the following experiments during the rebuttal process, where we compared with the model pre-trained on MolPCBA only and trained from scratch in downstream application, separately.
> > > >
> > > > |                                 | MolPCBA (↑) | ZINC(↓)    | Arxiv(↑)  |
> > > > |---------------------------------|-------------|------------|-----------|
> > > > | 1. pre-trained on all domains | **0.2916**  | **0.0613** | **72.30** |
> > > > | 2. pre-trained on MolPCBA          | 0.2914      | 0.0638     | 71.24     |
> > > > | 3. train-from-scratch              | 0.2910      | 0.0724     | 71.85     |
> > > >
> > > > We observe that
> > > > - **Pre-training is Crucial for Better Graph Representations (1, 2, 3)**: When comparing the performance of GFSE trained-from-scratch V.S. GFSE pre-trained on diverse domains, we observe that pre-training leads to better downstream performance in most cases. This underscores the importance of pre-training for generating more expressive and beneficial graph representations.
> > > >
> > > > - **GFSE essentially learns cross-domain structural patterns (1,3)**: Additionally, we observe that when GFSE is trained from scratch on a singular dataset, the downstream performance is consistently lower than when the model is pre-trained on multiple domains. This suggests that cross-domain pre-training is a crucial factor for improving performance.
> > > >
> > > > > Could you please explain how Llama+RRWP or Llama+GFSE is implemented? If possible, could you provide the code for this experiment?
> > > >
> > > > We follow the experimental setting in the paper (Line467-478) for the additional experiments. Specifically, we employ a lightweight MLP to align the PSE (i.e., RRWP or GFSE) with the language model’s embedding space. Then the projected embedding is prepended to original textual descriptions as a soft token. Thereby, Llama finetuning is conditioned on the graph structural information. We will release the complete codebase for all experiments once acceptance.

---

### Official Review · Reviewer_5q5Q · 2024-11-04

**Soundness:** 2
**Presentation:** 4
**Contribution:** 2
**Rating:** 6
**Confidence:** 5

**Summary:**

This paper introduces GFSE, a Graph Foundational Structural Encoder, designed to capture universal structural patterns in graph data, thus enabling effective cross-domain transfer. GFSE employs a Graph Transformer architecture with a focus on multiple structural pre-training objectives. By leveraging relative positional encoding and attention mechanisms, GFSE encodes complex topological information into a foundational model applicable to diverse domains, such as molecular and social networks. Experimental results reveal that GFSE significantly improves performance on various downstream tasks, including molecular property prediction and community detection.

**Strengths:**

1. The problem this paper aims to solve—Graph Foundation Model—is a highly relevant and challenging direction that has garnered significant attention.
2. The comparative experiments on SE and PE are extensive, thoroughly demonstrating the powerful capabilities of this work as a pre-trainable Structural Encoder.
3. The validation at the pre-training stage is highly meaningful. Compared to some studies that only evaluate downstream task performance, this pre-training stage validation provides deeper insights.

**Weaknesses:**

1. Although the title proposes to be ‘foundation model for graph structure encoding,’ the authors seem to aim to establish a connection with graph foundation models. However, the definition of ‘foundation model for graph structure encoding’ in the title remains unclear. GFM is expected to be pre-trainable on a wide range of graph data and applicable across various downstream tasks in different domains. In contrast, the GFSE in this paper is merely a pre-trainable positional encoding (PE) module. The subsequent integration with the downstream feature encoder is directly trained on downstream data, without pre-training on large-scale data or extracting transferable knowledge. Therefore, I find it difficult to consider this approach a GFM. Moreover, the experiments in the paper primarily compare various PE and SE methods. It seems more appropriate to position the scope of the paper as a pre-trained structural encoding model.
2. There are several aspects missing in the experimental validation. First, an important experiment is lacking: namely, an evaluation without pre-training the GFSE, where the full pipeline is applied directly to the downstream task (GFSE trained from scratch). This would allow for comparison with the pre-trained GFSE. The most similar experiment to this setup is in Table 4, but here the backbone models used in the ‘train from scratch’ and ‘fine-tuned’ modes are different, making a direct comparison infeasible. Second, in Table 5, the experiments combining GFSE with LLMs should be compared against existing models that integrate LLMs with graphs, as many of these models employ various methods for this integration (OFA [1] etc.).  Third, the dataset used for pre-training includes multiple collections, but the impact of the number of pre-training datasets on downstream performance is not shown. Additionally, is there a trend that shows better downstream performance as the number of pre-training datasets increases?
3. Many design choices lack motivation. For instance, regarding the selection of pre-training tasks, why were these four tasks chosen? In terms of task categories, the paper includes node-level and edge-level reconstruction tasks, as well as edge-level and graph-level contrastive learning tasks. Why not include graph-level reconstruction tasks or node-level contrastive learning tasks? Furthermore, why was motif counting chosen specifically for the node-level task instead of other reconstruction tasks? For the pre-training backbone, why were $P_M$ and $P_T$ input to the MLP separately rather than concatenated? Additionally, if only graph structure is being input, why rely solely on existing real-world graph data for pre-training rather than using some generated graph structures?
4. There are some details that need polishing. For example, in Figure 1, how should the textual features of B.2 be input to the Graph Foundational Structural Encoder? During the pre-training stage, the Graph Foundational Structural Encoder only receives $P$ and $R$ as inputs, without any text input. So, how are these textual features utilized in downstream tasks? Additionally, in Equation (1), $P$ and $R$ should have dimensions of $N \times (d + 1)$ and $N \times N \times (d + 1)$.


[1] One for All: Towards Training One Graph Model for All Classification Tasks

**Questions:**

See the weaknesses.

---

> ### Author Response · Authors · 2024-11-24
> **Response to reviewer 5q5Q (1/2)**
>
> Thanks for your valuable feedback and positive comments.  We address the potential concerns as follows.
>
> > Q1: The definition of ‘foundation model for graph structure encoding’ in the title remains unclear.
>
> We define a graph foundation model (GFM)  as "a model pretrained on extensive graph data, primed for adaptation across diverse downstream graph tasks.", which aligns with the concept proposed in the previous survey [1].  However, achieving a full-fledged GFM, capable of seamless adaptation across all domains, remains a significant challenge due to their heterogeneous features. In this work, we focus on a **different but complementary** goal—the exploration of universal structural patterns in graphs, which we believe are critical in developing a GFM. Our GFSE is not intended as an independent GFM, but rather as a foundational assistant to enhance the structural awareness of graph models, whether train-from-scratch models or pre-trained GFMs. Rather than positioning our work as a full-fledged GFM for diverse graph tasks, we aim to verify the effectiveness of structural patterns in improving the generic graph representation and its generalizability in cross-domain scenarios.
>
> [1] Towards Graph Foundation Models: A Survey and Beyond
>
> > Q2: An evaluation without pre-training the GFSE is missing. The impact of the number of pre-training datasets on downstream performance is not shown. Is there a trend that shows better downstream performance as the number of pre-training datasets increases?
>
> |                                 | MolPCBA (↑) | ZINC(↓)    | Arxiv(↑)  |
> |---------------------------------|-------------|------------|-----------|
> | 1. pre-trained on 100% all domains | **0.2916**  | **0.0613** | **72.30** |
> | 2. pre-trained on 50% all domains  | 0.2913      | 0.0658     | 72.13     |
> | 3. pre-trained on 10% all domains  | 0.2897      | 0.0703     | 71.76     |
> | 4. pre-trained on MolPCBA          | 0.2914      | 0.0638     | 71.24     |
> | 5. train-from-scratch              | 0.2910      | 0.0724     | 71.85     |
>
> Thanks for your suggestions. To address your concern, we have carefully analyzed the impact of the number of pre-training datasets on downstream performance, and the results are shown in the above table (the base model is GPS). Our findings show a few key trends:
> - **Performance Tends to Decrease as Pre-training Datasets Are Reduced (1-3)**: As the number of pre-training datasets decreases, we observe a slight reduction in downstream performance, indicating that a more diverse range of pre-training data can help improve the robustness of the model.
> - **Pre-training is Crucial for Better Graph Representations (1, 4, 5)**: When comparing the performance of GFSE trained-from-scratch V.S. GFSE pre-trained on different datasets, we observe that pre-training leads to better downstream performance in most cases. This underscores the importance of pre-training for generating more expressive and beneficial graph representations.
> - **Cross-domain Pre-training is Important (1,4)**: Additionally, we observe that when GFSE is pre-trained on a singular dataset (e.g., MolPCBA), the downstream performance is consistently lower than when the model is pre-trained on multiple domains. This suggests that cross-domain pre-training is a crucial factor for improving performance.
>
> We will include complete experimental results in our revised version to verify the above conclusion.
>
>
> >  Q3: The experiments combining GFSE with LLMs should be compared against existing models that integrate LLMs with graphs.
>
> We do compare our approach with GraphPEFT, one of the most recent SOTA models for text-attributed graphs, in Appendix F.2. Our GFSE consistently outperforms the version of GraphPEFT that skips the domain-specific pre-training step. Moreover, our model matches or even outperforms GraphPEFT which incorporates costly domain-specific pre-training. We acknowledge the benefit of evaluating against other models that integrate LLMs with graphs, and we are actively working on including more recent approaches in our revised version.

---

> > ### Author Response · Authors · 2024-11-24
> > **Response to reviewer 5q5Q (2/2)**
> >
> > > Q4: The chosen tasks lack motivation. Why is motif counting specifically chosen?
> >
> > The four tasks were chosen to cover a broad spectrum of graph structural patterns and to capture information at different levels of granularity. Each task targets a different aspect of graph encoding, as elaborated in **Section 3.2** and **Appendix B**. We also considered other potential reconstruction tasks, but motif counting was specifically chosen because it is a highly informative task that helps the model learn critical, recurrent subgraph structures that are fundamental to improving the model expressivity [2]. Regarding other potential tasks (e.g., graph-level reconstruction or node-level contrastive learning), we explored various combinations during the design phase. However, we found that these four tasks provided a balanced and effective mix for capturing both local and global structural information without introducing unnecessary complexity.
> >
> > [2] Improving graph neural network expressivity via subgraph isomorphism counting
> >
> > > Q5: Why $P_M$ and $P_T$ are input to the MLP separately, rather than concatenated? Why rely solely on existing real-world graph data for pre-training rather than using some generated graph structures?
> >
> > The proposed GFSE is built upon the GPS [3] framework, where the MPNN output and GlobalAttention output are added and input to the MLP (Equation (4) in [3]). This operation is also commonly used across other graph transformer models. Our primary motivation for using real-world graph data for pre-training was to reflect the diversity and complexity of real-world structures, which often feature varied node types, edge relationships, and topology. Moreover, we kindly bring to your consideration that our model, trained on real-world data, can also produce generic and powerful positional encodings that can be applied to **synthetic graph structures** (Table 1).
> >
> >  [3] Recipe for a General, Powerful, Scalable Graph Transformer
> >
> >
> > > Q6: Some details that need polishing.
> >
> > Thank you for bringing this to our attention. For text-attributed graphs, entities and edges are typically associated with textual descriptions. In downstream applications, these textual features in Figure 1(B.2) are processed by LLM through prompts, which are combined with the positional encoding generated by GFSE, allowing LLM to process both the graph structure and the textual descriptions simultaneously. For the shape of $P$ and $R$, we will revise the manuscript accordingly.

---

> ### Comment · Reviewer_5q5Q · 2024-11-25
>
> Thank you for your supplementary explanations and experiments during the rebuttal period, which addressed some of my concerns. I have raised my score to 6.
>
> However, my main concern remains the proposed pretraining structural encoder method, which seems to overclaim as a “foundation model for graph structure encoding.” I suggest narrowing the scope to a more appropriate range, similar to methods like GPS or “Graph Neural Networks with Learnable Structural and Positional Representations.”
>
> Additionally, if you claim this as a “foundation model for graph structure encoding,” it is essential to provide theoretical explanations about what transferable information the model learns and how this information enables broad improvements across diverse datasets. Furthermore, please clarify why the knowledge learned from these four tasks is sufficient to cover all transferable structural information. Why would adding more self-supervised tasks not further improve generalization?

---

> > ### Author Response · Authors · 2024-12-03
> >
> > Thanks for your follow-up feedback.
> > > Clarification of the "Foundation Model for Graph Structure Encoding" Claim
> >
> > We would like to emphasize that our approach represents an initial step toward developing a cross-domain graph structural encoder. While it may not yet embody the full potential of a “foundation model” as seen in other domains (e.g., NLP), we believe it captures universal structural patterns that can be leveraged across a wide variety of graph domains. However, we fully agree that a complete graph foundation model—in the sense of an extremely general, all-encompassing pre-trained model—is indeed a challenging and long-term goal. Our work contributes **a fundamental building block and pre-training strategy** toward this vision by proving the effectiveness of structural pre-training and transferable graph representations, which are crucial for future foundation models.
> >
> > > What transferable information does the model learn and how does this information enable broad improvements?
> >
> > We argue that the key to generalization across graph domains lies in the universal structural patterns encoded by our model.  As we have outlined in the paper (Line49-76), we aim to capture these structural patterns—such as community structures in social networks, hierarchical modularity in biological networks, and scale-free distributions in citation networks—because they are common but may hold different effects across a wide range of graph domains. These patterns act as the foundational building blocks for graph understanding and are transferable across downstream tasks. We design the graph contrastive learning task to deal with situations where the same pattern has different meanings in different domains. Simultaneously, other pre-training tasks capture the universality of these patterns across domains.
> >
> >
> > > Why Not Add More Self-Supervised Tasks?
> >
> > Adding more self-supervised tasks, in theory, might help refine the model further. However, the four tasks we selected are already sufficient for modeling universal graph attributes. They successfully recover most other positional and structure encodings (Table 11 in Appendix F), demonstrating **robust performance and a balanced capability to cover diverse structural information**. The key challenge in extending these tasks is that not all additional tasks would contribute meaningfully to improving the learned representations, especially given the **trade-off between task complexity and model efficiency**.

---

### Meta-Review · Area_Chair_vsoC · 2024-12-16

**Metareview:**

The reviewers agree that graph foundation model is a relevant and important topic to work on. Experimental validation, especially at the pre-training stage, is well executed and provided meaningful insights. The idea of a generalizable structural encoder is sound and represents a promising direction.

However, reviewers also find that there might be potential over-claiming of contributions in learning universal structure information on graphs, with some discrepancy between the stated positioning of the work and the actual scope of the experiments presented. Furthermore, the improvements in the results may be related to the selection of other hyperparameters, rather than the advances in the methodology itself.

**Additional Comments On Reviewer Discussion:**

While some reviewers raise their scores after rebuttal, the paper remains borderline. During the discussion phase, most of the reviewers (5q5Q, h6Am, jHor, 5nqo) expressed opinions not in favor of accepting this paper, mainly on the following issues:

1. Potential over claiming of the universal structure learning
2. Experimental improvement may be due to selection of hyperparameters rather than the advances in methodology.

---

### Decision · Program_Chairs · 2025-01-22

Reject